# First order expansion of convex regularized estimators

**Pierre C Bellec,**
Department of Statistics,
Rutgers University,
501 Hill Center,
Piscataway, NJ 08854, USA.
`pierre.bellec@rutgers.edu`

**Arun K Kuchibhotla,**
Department of Statistics,
The Wharton School,
University of Pennsylvania,
Philadelphia, PA 19104, USA.
`arunku@upenn.edu`

## Abstract

We consider first order expansions of convex penalized estimators in high-dimensional regression problems with random designs. Our setting includes linear regression and logistic regression as special cases. For a given penalty function $h$ and the corresponding penalized estimator $\hat{\beta}$, we construct a quantity $\eta$, the first order expansion of $\hat{\beta}$, such that the distance between $\hat{\beta}$ and $\eta$ is an order of magnitude smaller than the estimation error $\|\hat{\beta} - \beta^*\|$. In this sense, the first order expansion $\eta$ can be thought of as a generalization of influence functions from the mathematical statistics literature to regularized estimators in high-dimensions. Such first order expansion implies that the risk of $\hat{\beta}$ is asymptotically the same as the risk of $\eta$ which leads to a precise characterization of the MSE of $\hat{\beta}$; this characterization takes a particularly simple form for isotropic design. Such first order expansion also leads to inference results based on $\hat{\beta}$. We provide sufficient conditions for the existence of such first order expansion for three regularizers: the Lasso in its constrained form, the lasso in its penalized form, and the Group-Lasso. The results apply to general loss functions under some conditions and those conditions are satisfied for the squared loss in linear regression and for the logistic loss in the logistic model.

**Introduction.** We consider learning problems where one observes observations $(X_1, Y_1), ..., (X_n, Y_n)$ with responses $Y_i$ and feature vectors $X_i \in \mathbb{R}^p$. The literature of the past two decades has demonstrated the great success of regularized estimators that are commonly defined as solutions to regularized optimization problems of the form

$$\hat{\beta} = \operatorname{argmin}_{\beta \in \mathbb{R}^p} n^{-1} \sum_{i=1}^n \ell(Y_i, X_i^T \beta) + h(\beta), \tag{1}$$

where $\ell(\cdot, \cdot)$ is referred to as the loss (e.g. squared loss, logistic loss) and $h : \mathbb{R}^p \to \mathbb{R}$ is a regularization penalty (e.g. the $\ell_1$-norm for the Lasso, the $\ell_{2,1}$ norm for the Group-Lasso). All tuning parameters are included in $h(\cdot)$. The performance of such regularized estimators is measured in terms of prediction error or in terms estimation error $\|\hat{\beta} - \beta^*\|$ if the data comes from a model such as $Y = X\beta^* + \varepsilon$ for some noise random variable $\varepsilon$ in linear regression or

$$\mathbb{P}(Y = 1 | X = x) = 1/(1 + \exp(x^T \beta^*)) = 1 - \mathbb{P}(Y = 0 | X = x)$$

in logistic regression, where $\beta^*$ is the unknown coefficient vector. For instance, if $s = \|\beta^*\|_0$ is the sparsity of $\beta^*$ in the above model, and $(X_i, Y_i)_{i=1,...,n}$ are iid observations with the same distribution as $(X, Y)$, both the Lasso in linear regression and the logistic Lasso in logistic regression enjoy rate optimality: $\|\hat{\beta} - \beta^*\|^2 \le s \log(ep/s)/n$; see [35, 1] or the proof of Proposition 3.4 in Appendix F for self-contained proofs. The latter estimation bound is optimal in a minimax sense and cannot

be improved, and the minimax rate $s\log(ep/s)/n$ represents the scale below which uncertainty is unavoidable by information theoretic arguments, see for instance [36, Section 5].

We are interested in providing first order expansion of $\hat{\beta}$ at scales negligible compared to the minimax estimation rate, e.g. at scales negligible compared to $s\log(ep/s)/n$ in the aforementioned sparsity contexts. To be more precise, the results below will construct random first order expansion $\eta$ such that $\eta$ is measurable w.r.t. a much smaller sigma algebra than that generated by $(X_i, Y_i)_{i=1,\dots,n}$, and

$$\|\eta - \hat{\beta}\|_K^2 = o_p(1)\|\hat{\beta} - \beta^*\|_K^2 \quad \text{for some norm } \|\cdot\|_K \text{ related to the problem at hand,} \quad (2)$$

where $o_p(1)$ is a quantity that converges to 0 in probability. In other words, we provide a first-order expansion of $\hat{\beta}$ similar to an influence function expansion, cf. Section 1. This allows for understanding bias and standard deviation of $\hat{\beta}$ at a finer scale than simply showing that $\hat{\beta} - \beta^*$ converge to zero at the minimax rate. The present paper intends to answer the two questions below regarding such first order expansion.

- **(Q1)** How to construct $\eta$ such that (2) holds for a given convex regularized estimator such as (1)?

- **(Q2)** How are such first order expansions useful in high-dimensional learning problems where convex regularized estimators (1) are commonly used?

An expansion $\eta$ satisfying (2) is interesting in and by itself because it describes phenomena at a finer scale than most of the literature in high-dimensional problems which focuses on minimax prediction and estimation bounds. More importantly, we will see in Section 4 that such first-order expansions lead to exact identities for the loss of estimators, and in Section 5 that such first-order expansions can be used for inference (i.e., uncertainty quantification) about the unknown coefficient vector $\beta^*$.

**Notation.** Throughout the paper, $C_1, C_2, C_3, \dots$ denote positive absolute constants and we write $a \lesssim b$ if $a \leq Cb$ for some absolute constant $C > 0$. The Euclidean norm in $\mathbb{R}^p$ or in $\mathbb{R}^n$ is denoted by $\|\cdot\|$. For any positive definite matrix $A$, we write $\|u\|_A = \|A^{1/2}u\|$ for the matrix square-root $A$. For matrices, $\|\cdot\|_{op}$ and $\|\cdot\|_F$ denote the operator norm and Frobenius norm. For any real $a$, $a_+ = \max(0, a)$. If $S \subset \{1, \dots, p\}$, $v \in \mathbb{R}^p$, $M \in \mathbb{R}^{p \times p}$ then $v_S$ is the restriction $(v_j, j \in S)$ and $M_{S,S}$ is the square submatrix of $M$ made of entries indexed in $S \times S$.

# 1 Influence functions and Construction of $\eta$

To answer **(Q1)**, we start with a recap of unregularized estimators that correspond to $h(\cdot) \equiv 0$, when $p$ is fixed as $n \to +\infty$. In this case, it is well-known that certain smoothness assumptions on the loss such as twice differentiability [25, 19] or stochastic equicontinuity [42, 41] imply (for any norm, since all norms are equivalent in $\mathbb{R}^p$ for fixed $p$):

$$\left\| \hat{\beta} - \beta^* - \sum_{i=1}^n \frac{\psi(X_i, Y_i)}{n} \right\| = o_p(1)\|\hat{\beta} - \beta^*\| \Leftrightarrow (1 + o_p(1))\sqrt{n}(\hat{\beta} - \beta^*) = \sum_{i=1}^n \frac{\psi(X_i, Y_i)}{\sqrt{n}}, \quad (3)$$

for some target $\beta^*$ and a mean zero function $\psi(\cdot, \cdot)$ sometimes referred to as the influence function. See [25, Theorem 3.1], [42, Page 52], [41, Theorem 6.17], [19, Lemma 5.4] for details. In this case we can take $\eta = \beta^* + \sum \psi(X_i, Y_i)/n$ in (2). This representation allows us to claim asymptotic unbiasedness and fluctuations of order $n^{-1/2}$ for $\hat{\beta}$ around $\beta^*$. It also shows that estimator $\hat{\beta}$ behaves like an average and hence allows transfer of results (e.g., central limit theorems) for averages to study of $\hat{\beta}$ in terms of variance estimation, confidence intervals, hypothesis testing and bootstrap.

A general study of such representation for regularized problems is lacking in the literature. [23] is the first work that analyzed linear regression lasso when the number of covariates $p$ is fixed and does not change with the sample size $n$. In the more challenging regime where $p \geq n$, Theorem 5.1 of [22] provides a first order expansion allowing for $p$ to diverge (almost exponentially) with $n$. In the present work, we simplify and present a unified derivation of such first order expansion result, generalizing [22, Theorem 5.1] beyond the squared loss, beyond the $\ell_1$ penalty and beyond certain assumptions of [22] on $\mathbb{E}[X_i X_i^T]$. The derivation of (3) can be motivated by defining

$$\tilde{\eta} := \underset{\beta \in \mathbb{R}^p}{\operatorname{argmin}} \frac{1}{n}\sum_{i=1}^n \ell'(Y_i, X_i^\top \beta^*)X_i^\top(\beta - \beta^*) + \frac{1}{2n}\sum_{i=1}^n \ell''(Y_i, X_i^\top \beta^*)\{X_i^\top(\beta - \beta^*)\}^2 + h(\beta), \quad (4)$$

with $h(\cdot) \equiv 0$. Here and throughtout $\ell'(y, u)$ and $\ell''(y, u)$ represent (first and second) partial derivatives of $\ell$ with respect to $u$. The right hand side of (4) (with $h(\cdot) \equiv 0$) is the quadratic approximation of $\sum_{i=1}^n \ell(Y_i, X_i^\top \beta)/n$ around $\beta = \beta^*$ (without the term independent of $\beta$). The final first order expansion $\eta$ is obtained by replacing the quadratic part of the approximation by its expectation as in the next display. Following the intuitive construction of $\eta$ for the unregularized problem, we construct a first order expansion for the regularized problem as

$$\eta := \operatorname{argmin}_{\beta \in \mathbb{R}^p} \; n^{-1} \sum_{i=1}^n \ell'(Y_i, X_i^\top \beta^*) X_i^\top (\beta - \beta^*) + \tfrac{1}{2}(\beta - \beta^*)^\top K(\beta - \beta^*) + h(\beta)$$
$$:= \operatorname{argmin}_{\beta \in \mathbb{R}^p} \; \tfrac{1}{2} \left\| K^{1/2} \left( \beta - \beta^* - n^{-1} \sum_{i=1}^n K^{-1} X_i \ell'(Y_i, X_i^\top \beta^*) \right) \right\|^2 + h(\beta). \tag{5}$$

where $K := n^{-1} \sum_{i=1}^n \mathbb{E} \left[ \ell''(Y_i, X_i^\top \beta^*) X_i X_i^\top \right]$. From this definition, we can write $\eta = \eta_K \left( \beta^* + n^{-1} \sum_{i=1}^n K^{-1} X_i \ell'(Y_i, X_i^\top \beta^*) \right)$, for a function $\eta_K(\cdot)$ (depending on $h(\cdot), K$). Our main results prove under some mild assumptions that

$$\|\hat{\beta} - \eta_K(\beta^* + \tfrac{1}{n} \sum_{i=1}^n \ell'(Y_i, X_i^\top \beta^*) X_i)\|_K = o_p(1) \|\hat{\beta} - \beta^*\|_K.$$

Comparing this with (3) we note that for the unregularized problem, $\eta_K(\beta) = \beta$ is the identity.

## 2 Main Results: Approximation Theorem

We introduce the notion of Gaussian complexity for the following results. For any set $T \subset \mathbb{R}^p$ and a covariance matrix $\Sigma$, the Gaussian complexity of $T$ is given by

$$\gamma(T, \Sigma) := \mathbb{E} \left[ \sup_{u \in T: \|\Sigma^{1/2} u\| = 1} |g^\top \Sigma^{1/2} u| \right] = \mathbb{E} \left[ \sup_{u \in T} \frac{|g^\top \Sigma^{1/2} u|}{\|\Sigma^{1/2} u\|} \right], \tag{6}$$

where the expectation is with respect to the standard normal vector $g \sim N(0, I_p)$. We also need the notion of $L$-subGaussianity. A random vector $X$ is said to be $L$-subGaussian with respect to a (positive definite) matrix $\Sigma$ if

$$\forall u \in \mathbb{R}^p, \quad \mathbb{E}[\exp(u^T X)] \leq \exp(L^2 \|u\|_\Sigma^2 / 2) \qquad \text{where } \|u\|_\Sigma = \|\Sigma^{1/2} u\|. \tag{7}$$

This implies $\sup_u \mathbb{P}(|u^\top X| \geq t \|u\|_\Sigma) \leq 2 \exp(-t^2/(2L^2))$. Recall that the scaled norm $\|\cdot\|_K$ is defined by $\|u\|_K^2 = n^{-1} \sum_{i=1}^n \mathbb{E}[\ell''(Y_i, X_i^\top \beta^*)(X_i^\top u)^2]$. Consider the following assumptions:

**(A1)** There exists constants $0 \leq B, B_2, B_3 < \infty$ such that the loss satisfies $\forall u_1, u_2 \in \mathbb{R}, \forall y$,

$$\frac{|\ell''(y, u_1) - \ell''(y, u_2)|}{|u_1 - u_2|} \leq B, \qquad |\ell''(y, u_1)| \leq B_2, \qquad \sup_{u \in \mathbb{R}^p} \frac{\|\Sigma^{1/2} u\|^2}{\|K^{1/2} u\|^2} \leq B_3. \tag{8}$$

**(A2)** The observations $(X_1, Y_1), \ldots, (X_n, Y_n)$ are iid. Further $X_1, \ldots, X_n$ are mean zero and $L$-subGaussian with respect to their covariance $\Sigma$, i.e., (7) holds.

Note that $L$ in **(A2)** is necessarily no smaller than one, i.e., $L \geq 1$. Define the error

$$\mathcal{E} := \|\hat{\beta} - \beta^*\|_K + \|\eta - \beta^*\|_K \qquad \text{where} \qquad \|\cdot\|_K \text{ is the norm } \|u\|_K = \|K^{1/2} u\|. \tag{9}$$

The quantity $\mathcal{E}$ quantifies the error made by $\hat{\beta}$ and $\eta$ in estimating $\beta^*$ with respect to the norm $\|\cdot\|_K$. Bounds on $\|\hat{\beta} - \beta^*\|_K$ and $\|\eta - \beta^*\|_K$ follow from the existing literature; see [35] or Proposition 3.4 and its proof in Appendix F.

**Theorem 2.1.** *Let $r_n := n^{-1/2} \gamma(T, \Sigma)$ and assume that $r_n \leq 1$. Further assume (A1) and (A2) hold true. Then with probability at least $1 - 2e^{-C_4 n r_n^2} - 2e^{-C_5 \log n}$ we have the following:*

*1. If $\{\hat{\beta} - \beta^*, \eta - \beta^*\} \subseteq T$ then $\|\hat{\beta} - \eta\|_K \lesssim L B_2 B_3 r_n^{1/2} \mathcal{E} + B^{1/2}(B_3 L)^{3/2}(1 + r_n^3 \sqrt{n}) \mathcal{E}^{3/2}$.*

*2. If $\{\hat{\beta} - \eta, \hat{\beta} - \beta^*, \eta - \beta^*\} \subseteq T$ then $\|\hat{\beta} - \eta\|_K \lesssim B_2 B_3 L^2 r_n \mathcal{E} + B B_3^{3/2} L^3 (1 + r_n^3 \sqrt{n}) \mathcal{E}^2$.*

The set $T$ mentioned in Theorem 2.1(1) are available in the literature for many convex penalties. In the following, we will find this for constrained lasso, penalized lasso, and group lasso (with non-overlapping groups) under sharp conditions. We refer to [7] for slope penalty, and Negahban et al. [35, Lemma 1] and van de Geer [39, Def. 4.4 and Theorem 4.1] where set $T$ is presented for a general class of penalty functions including nuclear norm, group lasso (with overlapping groups).

Proofs of Theorem 2.1 and all following results are given in the supplement. An outline Theorem 2.1 is given in Section 6. Although Theorem 2.1 is stated under assumption **(A2)**, we present a deterministic version of the result (in Section 6) that replaces $r_n$ by suprema of different stochastic processes.

**Squared loss in the linear model.** Consider $\ell(y, u) = (y - u)^2/2$ and $n$ iid observations
$$Y_i = X_i^T \beta^* + \varepsilon_i, \text{ and } X_i \text{ is independent of } \varepsilon_i \text{ for } i = 1, \ldots, n, \tag{10}$$
Then we have $K = \Sigma = \mathbb{E}[X_1 X_1^T]$ and the second derivative $\ell''$ is constant. Hence condition (8) is satisfied with $B = 0$ and $B_2 = B_3 = 1$. The conclusions of the Theorem 2.1 can be rewritten as
$$\{\hat{\beta} - \beta^*, \eta - \beta^*\} \subseteq T \quad \Rightarrow \quad \|\hat{\beta} - \eta\|_K \lesssim L r_n^{1/2} \mathcal{E}. \tag{11}$$
$$\{\hat{\beta} - \eta, \hat{\beta} - \beta^*, \eta - \beta^*\} \subseteq T \quad \Rightarrow \quad \|\hat{\beta} - \eta\|_K \lesssim L^2 r_n \mathcal{E}, \tag{12}$$
where $\mathcal{E} = \|\Sigma^{1/2}(\hat{\beta} - \beta^*)\| + \|\Sigma^{1/2}(\eta - \beta^*)\|$. Since $r_n \leq 1$ (and typically $r_n \to 0$ while $L$ stays bounded, as we will see in the examples below), the inequality in (12) is stronger than the inequality in (11). In the linear model, we thus refer to inequality (11) as the "slow rate" inequality, and to (12) as the "fast rate" one. The set $T$ encodes the low-dimensional structure and characterizes the rate $r_n$ through the Gaussian complexity $\gamma(T, \Sigma)$. The fast rate inequality is granted provided that $T$ contains the difference $(\eta - \hat{\beta})$ additionally to the error vectors $\{\hat{\beta} - \beta^*, \eta - \beta^*\}$. Conditions that ensure the fast rate inequality will be made explicit in Section 3.2 for the Lasso.

**Logistic loss in the logistic model.** The following proposition shows that (8) is again satisfied.

**Proposition 2.2.** *Consider the logistic loss $\ell(y, u) = yu - \log(1 + e^u)$ for $y \in \{0, 1\}, u \in \mathbb{R}$. Assume that $(X_i, Y_i)_{i=1,\ldots,n}$ are iid satisfying the logistic regression model*
$$\mathbb{P}(Y_i = 1|X_i) = 1 - \mathbb{P}(Y_i = 0|X_i) = 1/(1 + \exp(X_i^\top \beta^*)),$$
*for some $\beta^* \in \mathbb{R}^p$ with $\|\Sigma^{1/2}\beta^*\| \leq 1$.[1] Assume (A2) holds. Then (8) holds with $B = 1/(6\sqrt{3})$, $B_2 = 1$ and an absolute constant $B_3 > 0$.*

In this logistic model, the conclusions of Theorem 2.1 present an extra term compared to the linear model with squared loss because the Lipschitz constant $B$ in (8) is non-zero: Theorem 2.1 reads that with high probability
$$\{\hat{\beta} - \beta^*, \eta - \beta^*\} \subset T \quad \Rightarrow \quad \|\hat{\beta} - \eta\|_K \lesssim L r_n^{1/2} \mathcal{E} + B^{1/2} L^{3/2} (1 + r_n^3 \sqrt{n}) \mathcal{E}^{3/2}, \tag{13}$$
$$\{\hat{\beta} - \eta, \hat{\beta} - \beta^*, \eta - \beta^*\} \subset T \quad \Rightarrow \quad \|\hat{\beta} - \eta\|_K \lesssim L^2 r_n \mathcal{E} + B L^3 (1 + r_n^3 \sqrt{n}) \mathcal{E}^2. \tag{14}$$

Similar to the case of squared loss, inequality (14) is stronger than inequality (13) when $\hat{\beta} - \eta$ belongs in $T$ additionally to $\{\hat{\beta} - \beta^*, \eta - \beta^*\} \subset T$.

## 3 What is the low-dimensional set $T$? Application to Lasso and Group-Lasso

We now provide applications of the above result to three different penalty functions commonly used in high-dimensional settings. Throughout this section, for any cone $T \subseteq \mathbb{R}^p$, let $\phi(T)$ be the smallest singular value of $\Sigma^{1/2}$ restricted to $T$, i.e., $\phi(T) = \min_{u \in T : \|u\| = 1} \|\Sigma^{1/2} u\|$. Further consider

**(N1)** The features are normalized such that $\Sigma_{jj} \leq 1$ for all $1 \leq j \leq p$.

### 3.1 Constrained Lasso

Let $R > 0$ be a fixed parameter. Our first example studies the constrained lasso penalty [38]
$$h(\beta) = +\infty \quad \text{if } \|\beta\|_1 > R \quad \text{and} \quad h(\beta) = 0 \quad \text{if } \|\beta\|_1 \leq R, \tag{15}$$
i.e., $h$ is the convex indicator function of the $\ell_1$-ball of radius $R > 0$. Applying the above result requires two ingredients: proving that the error vectors $\{\hat{\beta} - \beta^*, \eta - \beta^*\}$ belong to some set $T$ with high probability, and proving that $r_n = n^{-1/2} \gamma(T, \Sigma)$ is small. Define for any real $k \geq 1$,
$$T_{\texttt{lasso}}(k) := \{u \in \mathbb{R}^p : \|u\|_1 \leq \sqrt{k}\|u\|\}. \tag{16}$$
The parameter $k$ above will typically be a constant times $s = \|\beta^*\|_0$, the sparsity of $\beta^*$. If $R = \|\beta^*\|_1$, then the triangle inequality reveals that the error vectors of $\hat{\beta}$ and $\eta$ satisfy
$$\{\hat{\beta} - \beta^*, \eta - \beta^*\} \subseteq T := \{u \in \mathbb{R}^p : \|u_{S^c}\|_1 \leq \|u_S\|_1\}, \tag{17}$$
where $S = \{j = 1, \ldots, p : \beta_j^* \neq 0\}$ is the support of the true $\beta^*$ and $u_S$ is the restriction of $u$ to $S$. By the Cauchy-Schwarz inequality $\|u_S\|_1 \leq \sqrt{s}\|u_S\|_2$, thus $T$ in (17) satisfies $T \subset T_{\texttt{lasso}}(4s)$.

**Lemma 3.1.** *If (N1) holds and $k \geq 1$, then we have $\gamma(T, \Sigma) \lesssim \phi(T)^{-1}\sqrt{k \log(2p/k)}$ for any cone $T \subset T_{lasso}(k)$ where $T_{lasso}(k)$ is defined in* (16).

Hence under **(N1)** and by setting $k = 4s$ and $r_n = \phi(T)^{-1}\sqrt{s \log(ep/s)/n}$ we have in the linear model with squared loss that, with high probability,

$$\|\Sigma^{1/2}(\eta - \hat{\beta})\| \lesssim L\phi(T)^{-1/2}(s \log(ep/s)/n)^{1/4}(\|\Sigma^{1/2}(\hat{\beta} - \beta^*)\| + \|\Sigma^{1/2}(\eta - \beta^*)\|) \quad (18)$$

and we have established that $\eta$ is a first order expansion of $\hat{\beta}$ with respect to the norm $\|\cdot\|_\Sigma$ if $s \log(ep/s)/n \to 0$. It is informative to study the order of magnitude of the right hand side in (18). For that purpose, the following Lemma gives explicit bounds on $\|\Sigma^{1/2}(\hat{\beta} - \beta^*)\|$ and $\|\Sigma^{1/2}(\eta - \beta^*)\|$.

**Lemma 3.2.** *Consider the linear model with squared loss* (10) *and assume (A2). Let $\hat{\beta}, \eta$ in* (1) *and* (5) *with penalty* (15). *Then if $R = \|\beta^*\|_1$, we have with probability at least $1 - 2e^{-nr_n^2}$,*

$$\|\Sigma^{1/2}(\eta - \beta^*)\| \lesssim L\sigma^* r_n, \quad and \quad \|\Sigma^{1/2}(\hat{\beta} - \beta^*)\| \lesssim L\sigma^* r_n (1 - C_6 L^2 r_n)^{-1}, \quad (19)$$

*where $r_n = \phi(T)^{-1}\sqrt{s \log(ep/s)/n}$ and $(\sigma^*)^2 = (\varepsilon_1^2 + ... + \varepsilon_n^2)/n$.*

The above lemma provides a slight improvement in the rate compared to [17, Theorem 11.1(a)]. Combined with inequality (18), we have established that $\|\Sigma^{1/2}(\hat{\beta} - \eta)\| \lesssim L^2 \sigma^* r_n^{3/2}$. If $r_n \to 0$ (e.g., if $s \log(ep/s)/n \to 0$ while $\phi(T)$ stays bounded away from 0), this means that the distance $\|\Sigma^{1/2}(\hat{\beta} - \eta)\|$ between $\hat{\beta}$ and $\eta$ is an order of magnitude smaller than the risk bounds in (19).

Inclusion (17) is granted regardless of the loss $\ell$, as soon as $\beta^*$ lies on the boundary of $\{\beta \in \mathbb{R}^p : \|\beta\|_1 = R\}$. In logistic regression, i.e., the setting of Proposition 2.2 with the constrained Lasso penalty (15), inequality (13) yields that with high probability, $\|\eta - \hat{\beta}\|_K \lesssim L[r_n^{1/2} + L^{1/2}(1 + r_n^3\sqrt{n})\mathcal{E}^{1/2}]\mathcal{E}$. An extra term appears compared to the squared loss. In order to obtain a first-order expansion as in (2) requires $r_n \to 0$ as well as $(1 + r_n^3\sqrt{n})\mathcal{E}^{1/2} \to 0$. These conditions can be obtained if risk bounds such as (19) are available, see [35, 1] or Proposition 3.4 and its proof in Appendix F for applicable general techniques. A more detailed discussion of Logistic Lasso is given in the next subsection.

## 3.2 Penalized Lasso

We now consider the $\ell_1$-norm penalty

$$h(\beta) = \lambda\|\beta\|_1 \qquad \text{for some} \qquad \lambda \geq 0. \quad (20)$$

Here, the fact that $\hat{\beta} - \beta^*, \eta - \beta^* \in T$ for some low-dimensional cone $T$ is not granted almost surely, in that regard the situation differs from the constrained Lasso case in (17). We may find such low-dimensional cone $T$ simultaneously for $\hat{\beta}, \eta$ for both the squared loss and logistic loss as follows, using ideas from [35, 11]. Let $f_n$ be the convex function so that the objective in (1) is equal to $f_n(\beta) + h(\beta)$ and let $g_n$ be the convex function so that the objective in (5) is $g_n(\beta) + h(\beta)$. Since $\hat{\beta}$ and $\eta$ are solutions of the corresponding optimization problems (1) and (5),

$$\begin{aligned} h(\hat{\beta}) - h(\beta^*) &\leq f_n(\beta^*) - f_n(\hat{\beta}) \leq \nabla f_n(\beta^*)^T(\beta^* - \hat{\beta}), \\ h(\eta) - h(\beta^*) &\leq g_n(\beta^*) - g_n(\eta) \leq \nabla g_n(\beta^*)^T(\beta^* - \eta). \end{aligned} \quad (21)$$

Since $\nabla g_n(\beta^*) = \nabla f_n(\beta^*)$, both $\eta$ and $\hat{\beta}$ belong to the set $\hat{T} = \{b \in \mathbb{R}^p : h(b) - h(\beta^*) \leq \nabla f_n(\beta^*)^T(b - \beta^*)\}$. Next, for both the squared loss and the logistic loss, $\nabla f_n(\beta^*)$ has subGaussian coordinates under **(A2)**. Combining these remarks, we obtain the following, proved in supplement.

**Lemma 3.3.** *Let $h$ be as in* (20). *Consider the linear model* (10) *and assume (A2), (N1). Let $\xi > 0$ be a constant and let $\lambda = L\sigma^*(1 + 3\xi)\sqrt{2\log(p/s)/n}$ where $(\sigma^*)^2 = (\varepsilon_1^2 + \ldots + \varepsilon_n^2)/n$ and $\|\beta^*\|_0 = s$. Then*

$$\mathbb{P}\left[\{\hat{\beta} - \beta^*, \eta - \beta^*\} \subset T\right] \geq 1 - \frac{2}{\xi^2 \log(p/s)(p/s)^\xi} \quad where \quad T = T_{lasso}\left(s(6 + 2\xi^{-1})^2\right). \quad (22)$$

*If instead the logistic regression model and assumptions of Proposition 2.2 are fulfilled and $\lambda = (L/2)(1 + 3\xi)\sqrt{2\log(p/s)/n}$, then the previous display* (22) *also holds.*

The set $T$ above is the set $T_{\texttt{lasso}}(k)$ in (16) with $k = s(6 + 2\xi^{-1})^2$. Eq. (22) defines a low-dimensional cone $T$ that contains both error vectors $\hat{\beta} - \beta^*, \eta - \beta^*$ for the squared loss and the logistic loss. The Gaussian width of the set $T$ in (18) is already bounded in Lemma 3.1. Hence the Gaussian width of $T$ in the previous lemma is bounded from above as in the previous section, i.e., $\gamma(\Sigma, T) \lesssim \phi(T)^{-1}(6 + 2\xi^{-1})\sqrt{s \log(2p/s)}$ by Lemma 3.1, and the "slow rate" inequality (18) again holds with high probability, where $\phi(T)$ denotes the restricted eigenvalue of the set $T$ of the previous lemma. Risk bounds similar to (19) are given below. We emphasize here the fact that the error vectors of the Lasso belong to the cone (22) with high probability is not new: this is a powerful technique used throughout the literature on high-dimensional statistics starting from [11, 35]. The novelty of our results are inequalities such as (18) which shows that the distance $\|\Sigma^{1/2}(\hat{\beta} - \eta)\|$ is an order of magnitude faster than the minimax risk $\sqrt{s \log(ep/s)/n}$. We will now state a result similar to Lemma 3.2 for linear and logistic lasso.

**Proposition 3.4.** *Consider the penalized lasso estimator $\hat{\beta}$ given by*

$$\hat{\beta} := \text{argmin}_{\beta \in \mathbb{R}^p} \ \frac{1}{n} \sum_{i=1}^n \ell(Y_i, X_i^\top \beta) + \lambda \|\beta\|_1,$$

*where $\ell$ is either the squared or logistic loss and $\lambda$ is chosen as in Lemma 3.3 for some $\xi > 0$. Assume (A1), (A2). With $T$ defined in (22), assume that $\exists \theta > 0$ s.t. for all $u \in T$ with $\|u\|_K \leq 1$,*

$$\theta^2 \|u\|_K^2 \leq \frac{1}{n} \sum_{i=1}^n \left\{ \ell(Y_i, X_i^\top \beta^* + X_i^\top u) - \ell(Y_i, X_i^\top \beta^*) - u^\top X_i \ell'(Y_i, X_i^\top \beta^*) \right\}, \tag{23}$$

*as well as*

$$L(2 + 5\xi)\sqrt{2s \log(p/s)/n} \leq B_3^{1/2} \phi(T)\theta^2 \times \begin{cases} 1/\sigma^*, & \text{for } \ell, \text{ the squared loss,} \\ 2, & \text{for } \ell, \text{ the logistic loss.} \end{cases} \tag{24}$$

*Then with probability at least $1 - 2/(\xi^2 \log(p/s)(p/s)^\xi)$,*

$$\|\hat{\beta} - \beta^*\|_K \leq \frac{L(2 + 5\xi)}{B_3^{1/2} \phi(T)\theta^2} \sqrt{\frac{2s \log(p/s)}{n}} \times \begin{cases} \sigma^*, & \text{for } \ell, \text{ the squared loss,} \\ 0.5, & \text{for } \ell, \text{ the logistic loss.} \end{cases} \tag{25}$$

The proof is given Appendix F. Assumption (23) is the classical restricted strong convexity condition and we verify this for linear and logistic loss in Proposition F.1. Results similar to Proposition 3.4 are known in the literature [35] but the main novelty of our result is that the tuning parameter $\lambda$ is of order $\sqrt{\log(p/s)/n}$ and not $\sqrt{\log(p)/n}$ which proves the minimax optimal rate.

**Faster rates for the penalized lasso.** Fast rates for the Lasso can be obtained using the second inequality of Theorem 2.1, which when specialized to the squared loss gives (12). To verify the main additional assumption of $\hat{\beta} - \eta \in T$, we prove sparsity of $\eta$ and $\hat{\beta}$. Since $\hat{\beta}, \eta$ are defined through a penalized quadratic problem, we can leverage existing results in the literature that imply that $\eta, \hat{\beta}$ satisfies $\|\eta\|_0 \vee \|\hat{\beta}\|_0 \leq \tilde{C}s$ under suitable conditions on the design and as long as $s \log(ep/s)/n$ is small enough, for some constant $\tilde{C}$ that depends on the restricted singular values of $\Sigma$; cf., e.g.,[44, Lemma 1], [9, Theorem 3] [22, Lemma 3.5], [4, Lemma 6.1]. We prove such as result for the Group-Lasso in Proposition 3.7 below. Now we define the cones $T_0$ and $T$ as the sets

$$T_0 := \{u \in \mathbb{R}^p : \|u\|_0 \leq (2\tilde{C} + 1)s\} \subset T = \{u \in \mathbb{R}^p : \|u\|_1 \leq (2\tilde{C} + 1)^{1/2}\sqrt{s}\|u\|\}. \tag{26}$$

where the inclusion is obtained thanks to the Cauchy-Schwarz inequality. Then $\{\eta - \hat{\beta}, \hat{\beta} - \beta^*, \eta - \beta^*\} \subset T$ with high probability, the Gaussian width $\gamma(T, \Sigma)$ is bounded by Lemma 3.1 and the second inequality of Theorem 2.1 yields

$$\|\Sigma^{1/2}(\eta - \hat{\beta})\| \lesssim L^2 r_n \mathcal{E}, \quad \text{where} \quad r_n = \phi(T)^{-1}(s \log(ep/s)/n)^{1/2}.$$

Since $\mathcal{E} \lesssim r_n$ with high probability by known prediction bounds for the Lasso (see Proposition 3.4 and its proof in Appendix F for rates with squared and logistic loss), we obtain that with high probability,

$$\|\Sigma^{1/2}(\eta - \hat{\beta})\| \lesssim L^3 \phi^{-2}(T)s \log(ep/s)/n = L^3 r_n^2, \tag{27}$$

a rate that is the square of the minimax rate $r_n$, hence much smaller. For squared loss, this rate is also faster than the rate obtained in (18) which is of order $r_n^{3/2}$. This faster rate is obtained

thanks to the inclusion $\{\hat{\beta} - \eta, \hat{\beta} - \beta^*, \eta - \beta^*\} \subset T$, whereas in the setting of (18) we only had $\{\hat{\beta} - \beta^*, \eta - \beta^*\} \subset T$ but not $\hat{\beta} - \eta \in T$. To our knowledge, the only result in the literature similar to the above bounds is given by [22, Theorem 5.1]. This result from [22] shows that (27) holds for squared loss, provided that the covariance $\Sigma$ satisfies (a) the minimal singular value of $\Sigma$ is at least $c_3 > 0$, (b) the maximal singular value of $\Sigma$ is at most $c_4$, and (c) the covariance matrix $\Sigma$ satisfies

$$\max_{A \subset [p]:|A| \leq c_5 s} \max_{j \in A} \sum_{j \in A^c} |\Sigma_{ij}| \leq c_6. \tag{28}$$

Our results show that a first order expansion for the Lasso can be obtained using the slow rate bound (11) without the requirement that the spectral norm of $\Sigma$ is bounded, and for the fast rate without the stringent assumption (28) on the correlations of $\Sigma$. Not only do our results generalize Theorem 5.1 from [22] to more general $\Sigma$, Theorem 2.1 shows how to obtain first-order expansion $\eta$ beyond the squared loss (e.g. logistic loss) and beyond the $\ell_1$-penalty of the lasso: the previous subsection tackles the constrained Lasso penalty (15) and the next subsection tackles the Group-Lasso penalty.

Sparsity of $\eta$ for any general loss function is proved in Proposition 3.7. This alone does not imply inclusion of $\eta - \hat{\beta}$ in a low-dimensional set without sparsity of $\hat{\beta}$. Sparsity of $\hat{\beta}$ for general loss function is not well-studied but for logistic loss function Section D.4 of the supplement of [10] proves a sparsity bound of the form $\|\hat{\beta}\|_0 \leq \tilde{C}s$, similar to the squared loss. Unfortunately the proof there requires $\lambda \gtrsim \sqrt{\log p / n}$ instead of condition $\lambda \gtrsim \sqrt{\log(p/s)/n}$ used in Lemma 3.3 above and in [26, 37, 7, 4, 2].

### 3.3 Group-Lasso

Consider now a partition of $\{1, ..., p\}$ into $M$ groups $G_1, ..., G_M$. For simplicity, we assume that the groups have the same size $d = p/M$, which is typically the case in multitask learning with $d$ tasks and $M$ shared features. The Group-Lasso penalty studied in this subsection is

$$h(\hat{\beta}) = \lambda \sum_{k=1}^M \|\beta_{G_k}\| \qquad \text{where } \beta_{G_k} \in \mathbb{R}^{|G_k|} \text{ is the restriction } (\beta_j, j \in G_k). \tag{29}$$

In both the linear model with squared loss and in logistic regression with the logistic loss, we now show that $\hat{\beta} - \beta^*$ and $\eta - \beta^*$ belong to a low-dimensional cone (Lemma 3.5), and that the Gaussian width of this cone is bounded from above by $\sqrt{s}(\sqrt{d} + \sqrt{2 \log(M/s)})$ where $s$ is the number of groups with $\beta_{G_k}^* \neq 0$ (Lemma 3.6).

**Lemma 3.5.** *Consider the linear model (10) and assume that $\max_{k=1,...,M} \|\Sigma_{G_k,G_k}\|_{op} \leq 1$ and that each group has the same size $|G_k| = d = p/M$. Let $\xi > 0$ and set $\lambda = L\sigma^*(1 + \xi)[\sqrt{d} + (1 + 2\xi)\sqrt{2 \log(M/s)}]$ where $(\sigma^*)^2 = (\sum_{i=1}^n \varepsilon_i^2)/n$ and $s$ is the number of groups with $\beta_{G_k}^* \neq 0$. Then*

$$\mathbb{P}\left(\{\hat{\beta} - \beta^*, \eta - \beta^*\} \subset T\right) \geq 1 - 2/\left(2\xi^2 \log(M/s)(M/s)^\xi\right). \tag{30}$$

*for $T = \{\delta \in \mathbb{R}^p : \sum_{k=1}^M \|\delta_{G_k}\| \leq \sqrt{s}\|\delta\|_2(2 + 3\xi^{-1})\}$. If instead the logistic regression model and assumptions of Proposition 2.2 are fulfilled and $\lambda$ is as above with $\sigma^* = 1/2$, then (30) also holds.*

The fact that the Group-Lasso belongs with high probability to a low-dimensional cone has been used before to prove risk bounds, e.g., [31, 5]. However the tuning parameter in the above lemma is smaller than that used in these works and using such cones to prove first expansion as in the present paper are, to our knowledge, novel.

**Lemma 3.6.** *Assume that $\max_{k=1,...,M} \|\Sigma_{G_k,G_k}\|_{op} \leq 1$ and that each group has the same size $|G_k| = d = p/M$. The set $T$ defined in the previous lemma satisfies $\gamma(T, \Sigma) \lesssim C(\xi)\phi(T)^{-1}\sqrt{sd + s \log(M/s)}$ for some constant $C(\xi)$ that depends only on $\xi$.*

Hence if the number of groups $M$, the group-sparsity $s$ (number of groups such that $\beta_{G_k}^* \neq 0$) and the group size $d = p/M$ satisfy $(sd + s \log(M/s))/n \to 0$ while $\phi(T)$ is bounded away from 0, the above Lemmas combined with Theorem 2.1 imply that $\eta$ is a first-order expansion of $\hat{\beta}$ for both the squared loss in linear regression and logistic loss in the logistic model. We leverage this result to obtain an exact risk identity for the Group-Lasso in the next section.

**Proposition 3.7.** *Assume (A1), (A2). Let the setting of Lemma 3.6 be fulfilled. Fix $\lambda$ as in Lemma 3.5 for both squared and logistic loss for some $\xi > 0$ and $T$ be the cone defined in Lemma 3.5. If $\|K\|_{op} \leq C_{\max} < \infty$ and the assumptions of Proposition 3.4 hold, then*

$$\mathbb{P}\left(|\{k \in [M] : \eta_{G_k} \neq 0\}| \leq s\tilde{C}\right) \geq 1 - 2/(\xi^2 \log(M/s)(M/s)^\xi),$$

where $\tilde{C} := 1 + C_{\max}\{2(3+\xi)(1+\xi^{-1})\}^2 B_3^2 \phi(T)^{-2}$. *For the squared loss, the same holds for* $\hat{\beta}$ *with* $\tilde{C}$ *replaced by* $(1+o(1))\tilde{C}$ *provided* $\phi(T)^{-1}\sqrt{sd + s\log(M/s)}/\sqrt{n} \to 0$.

The proof is given in Appendix G. For the Lasso the assumption of $\|K\|_{op} \le C_{\max}$ can be relaxed to a bound on the sparse maximal eigenvalue of $K$ using devices from [44, Lemma 1], [47, Corollary 2], [8, Lemma 3] or [4, Proposition 7.4]. See also [31, Theorem 3.1] and [29, Lemma 6] for similar results for the Group-Lasso, although with a larger tuning parameter than in Proposition 3.7.

For the squared loss, if the condition number of $\Sigma$ stays bounded then $C_{\max}/\phi(T)^{-2}$ is also bounded. Then if $r_n = \sqrt{sd + s\log(M/s)}/\sqrt{n} \to 0$, Proposition 3.7 yields that both $\hat{\beta} - \eta$ belongs to the cone $\{\delta \in \mathbb{R}^p : \sum_{k=1}^M \|\delta_{G_k}\| \le (1+o(1))(2\tilde{C}s)^{1/2}\|\delta\|_2\}$, which yields the "fast rate" bound (12).

# 4 Application to exact risk identities

In the linear model with the squared loss and identity covariance ($\Sigma = I_p$), the expansion $\eta$ in (5) is particularly simple: $\eta$ becomes the proximal operator of the penalty $h$ at the point $z = \beta^* + n^{-1/2}\sum_{i=1}^n \varepsilon_i X_i$, i.e, $\eta = \mathsf{prox}_h(z)$ where $\mathsf{prox}_h(x) = \operatorname{argmin}_{b \in \mathbb{R}^p}\|x - b\|^2/2 + h(b)$. Hence the loss $\|\eta - \beta^*\|$ of $\eta$ has a simple form and if a first-order expansion (2) is available, for instance for the Lasso or Group-Lasso as a consequence of the Lemmas of the previous section, then the loss $\|\hat{\beta} - \beta^*\|$ is exactly the loss of $\mathsf{prox}(z)$ up to a smaller order term. Let us emphasize that the next result and following discussion provide exact risk identities for the loss $\|\hat{\beta} - \beta^*\|$ (as in (32) below), and not only upper bounds up to multiplicative constants.

**Theorem 4.1.** *[Exact Risk Identity] Consider the linear model* (10) *and the regularized problem* (1) *with an arbitrary proper convex function* $h(\cdot)$. *Assume that* $X_1, \ldots, X_n$ *are iid* $N(0, I_p)$ *independent of* $\varepsilon_1, \ldots, \varepsilon_n$ *and set* $\sigma^* = (\frac{1}{n}\sum_{i=1}^n \varepsilon_i^2)^{1/2}$. *Then with probability at least* $1 - 2\exp(-t^2/2)$,

$$\left| \|\hat{\beta} - \beta^*\| - \mathbb{E}_Z\left[\|\beta^* - \mathsf{prox}_h(\beta^* + n^{-1/2}\sigma^* Z)\|^2\right]^{1/2} \right| \le \frac{\sigma^*(t+1)}{n^{1/2}} + \|\hat{\beta} - \eta\| \qquad (31)$$

*where* $Z = \frac{1}{n^{1/2}\sigma^*}\sum_{i=1}^n \varepsilon_i X_i \sim N(0, I_p)$ *and* $\mathbb{E}_Z$ *denotes the expectation with respect* $Z$.

Theorem 4.1 is a generalization of Corollary 5.2 of [22] where the result is stated for $h(\beta) = \lambda\|\beta\|_1$ with $\lambda \gtrsim \sigma^*\sqrt{2\log(p)/n}$. For the case of lasso, either in its constrained form with tuning parameter chosen as in Lemma 3.3 or the penalized Lasso with tuning parameter as in Lemma 3.3, inequality (18) holds thanks to (17) and Lemma 3.1 for the constrained lasso, and thanks to Lemmas 3.1 and 3.3 for the penalized lasso. Hence for both the constrained and penalized lasso, if $\Sigma = I_p$ with Gaussian design, the second term on the right hand side of (31) is $O_p(\sigma^*/\sqrt{n}) + O_p(s\log(ep/s)/n)^{1/4})(\|\eta - \beta^*\| + \|\hat{\beta} - \beta^*\|)$. Hence if $s, n, p \to +\infty$ with $s\log(ep/s)/n \to 0$ and $s/p \to 0$ then (31) implies

$$\|\hat{\beta} - \beta^*\| = (1 + o_p(1))\mathbb{E}_Z[\|\beta^* - \mathsf{prox}_h(\beta^* + n^{-1/2}\sigma^* Z)\|^2]^{1/2}. \qquad (32)$$

For the penalized lasso, since $\eta$ represents a soft-thresholding operator which can be written in closed form, Theorem 4.1 allows a refined study of the risk of $\hat{\beta}$; see [14, Theorem 5.1]. Similarly for the group lasso, we have from Lemmas 3.5 and 3.6 that $\|\eta - \hat{\beta}\| = O_p((sd + s\log(M/s))^{1/4}/n^{1/4})\|\hat{\beta} - \beta^*\|$ (slow rate) which is again negligible relative to $\|\hat{\beta} - \beta^*\|$ if $(sd + s\log(M/s))/n \to 0, s/M \to 0$. Thus, (32) again holds true. For the group lasso $\eta = \mathsf{prox}_h(\beta^* + n^{-1/2}\sigma^* Z)$ represents the Block James-Stein estimator in the sequence model; see [13, Section 2.1].

Extending Corollary 5.2 of [22] to more general loss/penalty functions, the above device lets us characterize the risk $\|\hat{\beta} - \beta^*\|$: Up to a multiplicative constant of order $1 + o_p(1)$, the risk is the same as the risk of the proximal of $h$ in the Gaussian sequence model where one observes $N(\beta^*, (\sigma^*)^2/n)$.

# 5 Application to inference

The second application we wish to mention is related to confidence intervals in the linear model when the squared loss is used and $X_1, \ldots, X_n$ are iid Gaussian $N(0, \Sigma)$. Assume that one is interested in constructing a confidence interval for a specific linear combination $a^T\beta^*$ for some $a \in \mathbb{R}^p$.

Further assume, for simplicity, that $\Sigma$ is known and that $a$ is normalized with $\|\Sigma^{-1/2}a\| = 1$. Then previous works on *de-biasing* [45, 46, 20, 21, 40, 22, 4] suggests, given an estimator $\hat{\beta}$ that may be biased, to consider the bias-corrected estimator $\hat{\theta}$ defined by $\hat{\theta} = a^T\hat{\beta} + \|z_a\|^{-2}z_a^T(y - \mathsf{X}\hat{\beta})$, where $y = (Y_1, ..., Y_n)$ is the response vector and $\mathsf{X}$ is the design matrix with rows $X_1, ..., X_n$ and $z_a = \mathsf{X}\Sigma^{-1}a \sim N(0, I_n)$ is sometimes referred to as a score vector for the estimation of $a^T\beta^*$.

**Proposition 5.1.** *Assume that $X_1, ..., X_n$ are iid $N(0, \Sigma)$ and is independent of $\varepsilon = (\varepsilon_1, ..., \varepsilon_n) \sim N(0, I_n)$. Assume that for some cone $T$ and $r_n = \gamma(T, \Sigma)/\sqrt{n}$ we have*

$$\mathbb{P}(\|\Sigma^{1/2}(\hat{\beta} - \beta^*)\| + \|\Sigma^{1/2}(\eta - \beta^*)\| \leq C_7 r_n, \{\eta - \hat{\beta}, \eta - \beta^*, \hat{\beta} - \beta^*\} \subset T) \geq 1 - \alpha. \quad (33)$$

*Then for some $T_n$ with the t-distribution with $n$ degrees-of-freedom, with probability $1 - \alpha - 4e^{-nr_n^2/2}$,*

$$\sqrt{n}(\hat{\theta} - a^T\beta^*) - T_n = O_p((1 + r_n))\|\Sigma^{1/2}(\eta - \beta^*)\| + O_p(\sqrt{n}r_n)\|\Sigma^{1/2}(\eta - \hat{\beta})\|, \quad (34)$$

$$= O_p(r_n(1 + r_n)) + O_p(\sqrt{n}r_n^3). \quad (35)$$

Because $T_n$ has $t$ distribution with $n$ degrees of freedom, asymptotically $\mathbb{P}(|T_n| \leq 1.96) \to 0.95$ and hence from (35), we get that $\mathbb{P}(n^{1/2}|\hat{\theta} - a^\top\beta^*| \leq 1.96) \to 0.95$ if $r_n^3\sqrt{n} \to 0$. Therefore, $[\hat{\theta} - 1.96/n^{1/2}, \hat{\theta} + 1.96/n^{1/2}]$ represents a 95% confidence interval for $a^\top\beta^*$. Conclusion (34) is a consequence of Theorem 2.1.

**Lasso.** Eq. (33) is satisfied for the penalized Lasso for $r_n = \sqrt{s\log(ep/s)/n}$ and the cone $T$ in (26), in situations where $\|\hat{\beta}\|_0 \leq \tilde{C}s$ with high probability as explained in the discussion surrounding (26). In order to construct confidence interval based on (34), the right hand side of (35) needs to converges to 0. This is the case if $r_n \to 0$ and $\sqrt{n}r_n^3 \to 0$. For the Lasso with $r_n = s\log(ep/s)/n$, this translates to the sparsity condition $s^3\log(ep/s)^3/n^2 \to 0$, i.e., $s = o(n^{2/3})$ up to logarithmic factors. Hence the first order expansion results of the present paper lets us derive de-biasing results for the Lasso beyond the condition $s \lesssim \sqrt{n}$ required in the early results [46, 20, 40] on de-biasing (other recent approaches, [22, 4] also allow to prove such result beyond $s \lesssim \sqrt{n}$). Moreover, the above proposition is general and apply to any regularized estimator such that (33) holds, with suitable bounds on the Gaussian complexity $\gamma(T, \Sigma)$. For $s \ggg n^{2/3}$, the estimator $\hat{\theta}$ requires an adjustment for asymptotic normality in the form a degree-of-freedom adjustment [4].

**Group-Lasso.** If $s$ is the number of non-zero groups, $r_n = \sqrt{sd + s\log(M/s)}/\sqrt{n}$ and the condition number of $\Sigma$ is bounded, then (33) holds thanks to Proposition 3.7, the last paragraph of Section 3.3 and the risk bound (86). Here, (35) is $o(1)$ if and only if $(sd + s\log(M/s))/n^{2/3} \to 0$. This improves the sample size requirement of [34], although $\Sigma$ is assumed known in Proposition 5.1.

# 6 Proof sketch of Theorem 2.1 (Detailed proofs are given in Appendix E)

**Theorem 6.1.** *Define $\hat{K} := n^{-1}\sum_{i=1}^n \ell''(Y_i, X_i^\top\beta^*)X_iX_i^\top$. Under assumption (A1), we have*

*(i) If $\{\hat{\beta} - \beta^*, \eta - \beta^*\} \subseteq T$ then $\|\hat{\beta} - \eta\|_K \lesssim Q_{n,1}^{1/2}\mathcal{E} + B^{1/2}Z_n^{1/2}\mathcal{E}^{3/2}$.*

*(ii) If $\{\hat{\beta} - \eta, \hat{\beta} - \beta^*, \eta - \beta^*\} \subseteq T$ then $\|\hat{\beta} - \eta\|_K \lesssim Q_{n,2}\mathcal{E} + BZ_n\mathcal{E}^2$,*

*where*

$$Q_{n,1} = \sup_{u \in T}\left|\frac{u^\top\hat{K}u}{\|u\|_K^2} - 1\right|, \quad Q_{n,2} = \sup_{u,v \in T}\frac{|u^\top(\hat{K} - K)v|}{\|u\|_K\|v\|_K} \quad and \quad Z_n = \sup_{u \in T}\frac{1}{n}\sum_{i=1}^n\frac{|X_i^\top u|^3}{\|u\|_K^3}.$$

Theorem 6.1 follows from the strong convexity of the objective function of $\eta$ with respect to the norm $\|\cdot\|_K$ (cf. for instance, Lemma 1 of [6]) combined with Taylor expansions of the loss $\ell$. Next, to prove Theorem 2.1, it remains to bound $Q_{n,1}(T), Q_{n,2}(T)$ and $Z_n(T)$. The quadratic processes $Q_{n,1}(T), Q_{n,2}(T)$ and cubic process $Z_n(T)$ can be bounded in terms of $\gamma(T, \Sigma)$ using generic chaining results, Theorem 1.13 of Mendelson [33] and Eq. (3.9) of [32], as follows.

**Proposition 6.2.** *[Control of $Q_{n,1}, Q_{n,2}$ and $Z_n$] Under assumptions (A1) and (A2), we have*

*(i) With probability $1 - 2\exp(-C_8t^2\gamma^2(T, \Sigma))$,*

$$\max\{Q_{n,1}(T), Q_{n,2}(T)\} \leq C_9B_2B_3L^2\left(tn^{-1/2}\gamma(T, \Sigma) + t^2n^{-1}\gamma^2(T, \Sigma)\right).$$

*(ii) With probability $1 - 2\exp(-C_{10}t\log n)$, $\quad Z_n(T) \leq C_{11}B_3^{3/2}L^3\left(1 + n^{-1}\gamma^3(T, \Sigma)\right)t^3$.*

## Footnotes

[1]The constant 1 can be replaced by another absolute constant; this will only change $B_3$ to a different constant.

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
