[Supplementary Material · neurips_2019.pdf]

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

## SUPPLEMENT

| | Lasso | Group-Lasso, $M$ groups of size $d = p/M$ |
|---|---|---|
| Tuning parameter | $\lambda \gtrsim [\frac{2}{n} \log \frac{p}{s}]^{\frac{1}{2}}$ in (20) | $\lambda \gtrsim [d + \frac{2}{n} \log \frac{M}{s}]^{\frac{1}{2}}$ in (29) |
| Minimax rate $r_n$ | $\|\hat{\beta} - \beta^*\|_\Sigma \lesssim r_n$ $r_n = [\frac{2s}{n} \log \frac{p}{s}]^{\frac{1}{2}}$ | $\|\hat{\beta} - \beta^*\|_\Sigma \lesssim r_n$ $[\frac{d}{n} + \frac{s}{n} \log \frac{M}{s}]^{\frac{1}{2}}$ |
| Gaussian width bound | $\gamma(T, \Sigma) \lesssim [s \log \frac{p}{s}]^{1/2}$ | $\gamma(T, \Sigma) \lesssim [sd + s \log \frac{M}{s}]^{1/2}$ by Lemma 3.6 |
| Restricted Eigenvalue (RE) | $\|\eta - \hat{\beta}\|_\Sigma \lesssim r_n^{3/2}$ by (11) and Lemma 3.3 (Lasso) or Lemma 3.5 (GL) | |
| $\|\Sigma\|_{op} \vee \|\Sigma^{-1}\|_{op} \leq C$ | $\|\eta - \hat{\beta}\|_\Sigma \lesssim r_n^2$ by (12), Proposition 3.7 | |

Table 1: Summary of rates for $\|\hat{\beta} - \beta^*\|_\Sigma$ and $\|\eta - \hat{\beta}\|_\Sigma$ for the squared loss. For the Lasso, $s$ is the sparsity of $\beta^*$ while for the Group-Lasso $s$ is the number of non-zero groups in $\beta^*$. If $r_n \to 0$ then $\|\eta - \hat{\beta}\|_\Sigma$ is an order of magnitude smaller than $\|\hat{\beta} - \beta^*\|_\Sigma$ and the minimax rate. In this table $\gtrsim$ may hide constants depending on the subgaussian parameter $L$ as well as restricted eigenvalues of $\Sigma$, denoted by $\phi(T)$ in the paper.

| | Lasso | Group-Lasso, $M$ groups of size $d = \frac{p}{M}$ |
|---|---|---|
| Tuning parameter | $\lambda \gtrsim [\frac{2}{n} \log \frac{p}{s}]^{\frac{1}{2}}$ in (20) | $\lambda \gtrsim [d + \frac{2}{n} \log \frac{M}{s}]^{\frac{1}{2}}$ in (29) |
| Minimax rate $r_n$ | $\|\hat{\beta} - \beta^*\|_K \lesssim r_n$ $r_n = [\frac{2s}{n} \log \frac{p}{s}]^{\frac{1}{2}}$ (Prop. 3.4) | $\|\hat{\beta} - \beta^*\|_K \lesssim r_n$ $[\frac{d}{n} + \frac{s}{n} \log \frac{M}{s}]^{\frac{1}{2}}$ |
| Gaussian width bound | $\gamma(T, \Sigma) \lesssim [s \log \frac{p}{s}]^{1/2}$ | $\gamma(T, \Sigma) \lesssim [sd + s \log \frac{M}{s}]^{1/2}$ by Lemma 3.6 |
| $RSC$ (cf. Appendix F) | $\|\eta - \hat{\beta}\|_K \lesssim r_n^{3/2}(1 + r_n^3 \sqrt{n})$ by (13) and Lemma 3.3 or Lemma 3.5 | |
| $\|K\|_{op} \vee \|K^{-1}\|_{op} \leq C$ | $\|\eta - \hat{\beta}\|_K \lesssim r_n^2(1 + r_n^3 \sqrt{n})$ by (14) and Proposition 3.7 | |

Table 2: Summary of rates for $\|\hat{\beta} - \beta^*\|_K$ and $\|\eta - \hat{\beta}\|_K$ for the logistic loss. For the Lasso, $s$ is the sparsity of $\beta^*$ while for the Group-Lasso $s$ is the number of non-zero groups in $\beta^*$. If $r_n \to 0$ as well as $r_n^3 \sqrt{n} \to 0$ then $\|\eta - \hat{\beta}\|_K$ is an order of magnitude smaller than $\|\hat{\beta} - \beta^*\|_K$ and the minimax rate. In this table $\gtrsim$ may hide constants depending on the subgaussian parameter $L$, the constants $B_3$ and Restricted Strong Convexity (RSC) constants.

**Proofs.** All Theorems, Lemmas and Propositions from the submission are proved in the present supplement. The results are restated before their proofs for convenience.

## A Proofs of Section 3

**Lemma 3.1.** *If (N1) holds and $k \geq 1$, then we have $\gamma(T, \Sigma) \lesssim \phi(T)^{-1} \sqrt{k \log(2p/k)}$ for any cone $T \subset T_{lasso}(k)$ where $T_{lasso}(k)$ is defined in* (16).

*Proof of Lemma 3.1.* This is a consequence of Lemma 3.6 proved below, by taking $p$ groups of size $d = 1$, i.e., the groups are $G_j = \{j\}$ for each $j = 1, ..., p$ and $M = p$. The condition $\max_{k=1,...,M} \|\Sigma_{G_k, G_k}\|_{op} \leq 1$ necessary to apply Lemma 3.5 is equivalent to the normalization **(N1)**. $\square$

**Lemma 3.2.** *Consider the linear model with squared loss* (10) *and assume (A2). Let $\hat{\beta}, \eta$ in* (1) *and* (5) *with penalty* (15). *Then if $R = \|\beta^*\|_1$, we have with probability at least $1 - 2e^{-nr_n^2}$,*

$$\|\Sigma^{1/2}(\eta - \beta^*)\| \lesssim L\sigma^* r_n, \quad and \quad \|\Sigma^{1/2}(\hat{\beta} - \beta^*)\| \lesssim L\sigma^* r_n (1 - C_{12} L^2 r_n)^{-1}, \qquad (19)$$

*where $r_n = \phi(T)^{-1} \sqrt{s \log(ep/s)/n}$ and $(\sigma^*)^2 = (\varepsilon_1^2 + ... + \varepsilon_n^2)/n$.*

*Proof of Lemma 3.2.* Since $R = \|\beta^*\|_1$, the inclusion (17) holds by the triangle inequality, i.e., we have $\hat{\beta} - \beta^* \in T$ as well as $\eta - \beta^* \in T$.

Next, we first bound the loss of $\eta$. The optimization problem (5) for the squared loss for the penalty (15) can be rewritten as

$$\eta = \operatorname*{argmin}_{\beta \in \mathbb{R}^p : \|\beta\|_1 \leq R} \frac{1}{2} \|\Sigma^{1/2}(\beta - \beta^*) - n^{-1/2}Z\|^2, \qquad \text{where} \qquad Z = \frac{1}{\sqrt{n}} \sum_{i=1}^{n} \varepsilon_i \Sigma^{-1/2} X_i.$$

By optimality of $\eta$ for the above optimization problem, we have (see, e.g., the properties of convex projections in [3]) that

$$\|\Sigma^{1/2}(\eta - \beta^*)\| \leq \frac{1}{\sqrt{n}} \frac{(\Sigma^{1/2}(\eta - \beta^*))^T Z}{\|\Sigma^{1/2}(\eta - \beta^*)\|} \leq \frac{\sigma^*}{\sqrt{n}} \sup_{u \in T : \|\Sigma^{1/2}u\|=1} u^T \Sigma^{1/2} Z / \sigma^*.$$

Next, notice that $Z/\sigma^*$ is $L$-subgaussian because for any $u \in \mathbb{R}^p$, by independence,

$$\mathbb{E}[\exp(u^T Z)] = \prod_{i=1}^{n} \mathbb{E}[e^{n^{-1/2}\varepsilon_i X_i^T \Sigma^{-1/2} u}] \leq \prod_{i=1}^{n} e^{n^{-1}\varepsilon_i^2 L^2 \|u\|^2/2} = e^{(\sigma^*)^2 L^2 \|u\|^2/2}. \qquad (36)$$

By a tail bound on suprema of subGaussian processes, we obtain that with probability at least $1 - e^{-t^2}$, inequality $\sup_{u \in T : \|\Sigma^{1/2}u\|=1} u^T \Sigma^{1/2} Z / \sigma^* \leq C_{13}(\gamma(T, \Sigma) + t)$ holds for some absolute constant $C_{13}$. We have proved in Lemma 3.1 that $\gamma(T, \Sigma) \leq C_{14}\phi(T)^{-1}\sqrt{s \log(2p/s)}$ for another absolute constant. The choice $t = \phi(T)^{-1}\sqrt{s \log(2p/s)} = r_n \sqrt{n}$ completes the proof for $\eta$.

We now prove the bound for $\hat{\beta}$. For the squared loss in the linear model,

$$\hat{\beta} = \operatorname*{argmin}_{\beta \in \mathbb{R}^p : \|\beta\|_1 \leq R} \|\mathsf{X}(\beta - \beta^*) - \varepsilon\|^2/(2n)$$

where $\mathsf{X}$ is the design matrix with rows $X_1, ..., X_n$ and $\varepsilon = (\varepsilon_1, ..., \varepsilon_n)$. The optimality conditions of the above optimization problem yields that

$$\frac{\|\mathsf{X}(\hat{\beta} - \beta^*)\|^2}{n\|\Sigma^{1/2}(\hat{\beta} - \beta^*)\|} \leq \frac{\varepsilon^T \mathsf{X}(\hat{\beta} - \beta^*)}{n\|\Sigma^{1/2}(\hat{\beta} - \beta^*)\|} = \frac{1}{\sqrt{n}} \frac{(\Sigma^{1/2}(\hat{\beta} - \beta^*))^T Z}{\|\Sigma^{1/2}(\hat{\beta} - \beta^*)\|} \leq \frac{\sigma^*}{\sqrt{n}} \sup_{u \in T : \|\Sigma^{1/2}u\|=1} \frac{u^T \Sigma^{1/2} Z}{\sigma^*}.$$

We have already bounded in the previous paragraph the supremum in the right hand side with probability at least $1 - e^{-nr_n^2}$. It remains to show that the left hand side is larger than $\|\Sigma^{1/2}(\hat{\beta} - \beta^*)\|(1 - C_{15}r_n)$ with high probability. Since $\hat{\beta} - \beta^* \in T$, an application of [28] to the set $(\Sigma^{1/2}T) \cap \{v \in \mathbb{R}^p : \|v\| = 1\}$ yields that, with probability at least $1 - 2e^{-r_n^2 n}$,

$$\left| \frac{\|\mathsf{X}(\hat{\beta} - \beta^*)\|}{\|\Sigma^{1/2}(\hat{\beta} - \beta^*)\|} - \sqrt{n} \right| \leq C_{16}(\gamma(T, \Sigma) + \sqrt{n}r_n) \leq C_{17}\sqrt{n}r_n.$$

In the same event, we have $\|\mathsf{X}(\hat{\beta} - \beta^*)\|^2/n \geq (1 - C_{18}r_n)^2\|\Sigma^{1/2}(\hat{\beta} - \beta^*)\|^2$ and the proof is complete. $\square$

The following Lemma will be useful.

**Lemma A.1.** *The following, (i) in the linear model and (ii) in the logistic model, hold for any convex penalty $h$.*

*(i) Consider the linear model (10), assume (A2) and assume that $(\varepsilon_1, ..., \varepsilon_n)$ is independent of $(X_1, ..., X_n)$ and let $(\sigma^*)^2 = (1/n)\sum_{i=1}^{n} \varepsilon_i^2$. Then almost surely,*

$$\{\eta, \hat{\beta}\} \subset \hat{T} = \{b \in \mathbb{R}^p : \sqrt{n}(h(b) - h(\beta^*)) \leq Z^T \Sigma^{1/2}(b - \beta^*)\} \qquad (37)$$

*where $Z$ is an $L(\sigma^*)$-subgaussian vector in the sense that $\mathbb{E}\exp(u^T Z) \leq \exp(L^2(\sigma^*)^2\|u\|^2/2)$.*

*(ii) Consider the logistic model and assume (A2). Then almost surely*

$$\{\eta, \hat{\beta}\} \subset \tilde{T} = \{b \in \mathbb{R}^p : \sqrt{n}(h(b) - h(\beta^*)) \leq \tilde{Z}^T \Sigma^{1/2}(b - \beta^*)\}$$

*where $\tilde{Z}$ is an $L/2$-subgaussian vector in the sense that $\mathbb{E}\exp(u^T \tilde{Z}) \leq \exp((L/2)^2\|u\|^2/2)$.*

*Proof.* (i) We first prove the result in the linear model for the squared loss. Here (21) holds with $g_n$ and $f_n$ defined before (21), so that

$$\sqrt{n}\nabla f_n(\beta^*) = \sqrt{n}\nabla g_n(\beta^*) = \Sigma^{1/2}Z \qquad \text{where} \qquad Z \triangleq \frac{1}{\sqrt{n}}\sum_{i=1}^{n}\varepsilon_i\Sigma^{-1/2}X_i \qquad (38)$$

in the linear model for the squared loss. Let $\hat{T} = \{b \in \mathbb{R}^p : h(b) - h(\beta^*) \le Z^T\Sigma^{1/2}(b - \beta^*)\}$. Then both $\eta$ and $\hat{\beta}$ belong to $\hat{T}$ by (21). We already proved that $Z$ is is $L\sigma^*$-subgaussian in the sense that $\mathbb{E}[\exp(Z^Tu)] \le \exp(L^2(\sigma^*)^2\|u\|^2/2)$ for all $u \in \mathbb{R}^p$ in (36); this completes the proof of (i).

(ii) In logistic regression with the logistic loss, (21) again, holds, i.e., $\{\eta, \hat{\beta}\}$ belong to $\tilde{T} = \{b \in \mathbb{R}^p : h(b) - h(\beta^*)) \le \tilde{Z}^T\Sigma^{1/2}(b - \beta^*)\}$ where

$$\tilde{Z} \triangleq \sqrt{n}\Sigma^{-1/2}\nabla f_n(\beta^*) = \sqrt{n}\Sigma^{-1/2}\nabla g_n(\beta^*) = \frac{1}{\sqrt{n}}\sum_{i=1}^{n}\left(Y_i - \frac{1}{1 + e^{X_i^T\beta^*}}\right)\Sigma^{-1/2}X_i \quad (39)$$

where $g_n$ and $f_n$ are defined before (21). We now show that $\tilde{Z} \in \mathbb{R}^p$ is a subgaussian vector. Note that $\mathbb{E}[Y_i|X_i] = 1/(1 + e^{X_i^T\beta^*})$ so that $\mathbb{E}[\tilde{Z}|X_1, ..., X_n] = 0$ and $\mathbb{E}[\tilde{Z}] = 0$. If $B$ is Bernoulli with parameter $p$, then $\mathbb{E}[e^{t(B-p)}] = pe^{t(1-p)} + (1-p)e^{t(-p)}$ which is maximized at $p = 1/2$, hence $\mathbb{E}[e^{t(B-p)}] \le (e^{t/2} + e^{-t/2})/2$. For any $u \in \mathbb{R}^p$, set $v = \Sigma^{-1/2}u$ and notice that by independence and the law of total expectation,

$$\mathbb{E}[e^{\tilde{Z}^Tu}] = \prod_{i=1}^{n}\mathbb{E}[e^{(Y_i - \mathbb{E}[Y_i|X_i])\frac{X_i^Tv}{\sqrt{n}}}] \le \prod_{i=1}^{n}\frac{\mathbb{E}e^{\frac{X_i^Tv}{2\sqrt{n}}} + \mathbb{E}e^{\frac{-X_i^Tv}{2\sqrt{n}}}}{2} \le e^{L^2\|\Sigma^{1/2}v\|^2/8} = e^{L^2\|u\|^2/8},$$

where for the last inequality we use that each $X_i$ is $L$-subgaussian. Hence $\tilde{Z}$ is $L/2$-subgaussian for the logistic loss. $\qquad\square$

**Lemma 3.3.** *Let $h$ be as in* (20). *Consider the linear model* (10) *and assume (A2), (N1). Let $\xi > 0$ be a constant and let $\lambda = L\sigma^*(1 + 3\xi)\sqrt{2\log(p/s)/n}$ where $(\sigma^*)^2 = (\varepsilon_1^2 + ... + \varepsilon_n^2)/n$ and $\|\beta^*\|_0 = s$. Then*

$$\mathbb{P}\left[\{\hat{\beta} - \beta^*, \eta - \beta^*\} \subset T\right] \ge 1 - \frac{2}{\xi^2\log(p/s)(p/s)^\xi} \text{ where } T = T_{lasso}\left(s(6 + 2\xi^{-1})^2\right). \quad (22)$$

*If instead the logistic regression model and assumptions of Proposition 2.2 are fulfilled and $\lambda = (L/2)(1 + 3\xi)\sqrt{2\log(p/s)/n}$, then the previous display* (22) *also holds.*

*Proof of Lemma 3.3.* We will apply the previous lemma, but first let us derive some properties of subgaussian vectors.

Let $U$ be a random vector valued in $\mathbb{R}^p$ such that each component $U_j$ is 1-subgaussian in the sense that $\mathbb{E}[e^{tU_j^2}] \le e^{t^2/2}$, for each $j = 1, ..., p$. Let $\mu > 0$ be a deterministic real.

Since $U_j$ is 1-subgaussian, $\mathbb{P}(U_j > \sqrt{2x}) \le e^{-x}$ by a Chernoff bound, hence $(U_j)_+^2/2$ is stochastically dominated by an exponential random variable $\tau$ with parameter 1 and there exists a probability space on which both $U_j$ and $\tau$ are defined such that $U_j \le \sqrt{2\tau}$ holds almost surely. Hence using $|\sqrt{a} - \sqrt{b}|^2 \le a^2 - b^2$ for any $a > b > 0$,

$$\mathbb{E}[(U_j - \mu)_+^2] \le \mathbb{E}[(\sqrt{2\tau} - \mu)_+^2] \le \int_{\mu^2/2}^{\infty}(\sqrt{2t} - \mu)_+^2 e^{-t}dt \le 2\int_{\mu^2/2}^{\infty}(t - \mu^2/2)_+e^{-t}dt = 2e^{-\mu^2/2}.$$

The same holds with $U_j$ replaced by $-U_j$. For $\mu = (1 + \xi)\sqrt{2\log(p/s)}$ for $\xi \ge 0$, this shows that

$$\mathbb{E}\sum_{j=1}^{p}(|U_j| - \mu)_+^2 \le \mathbb{E}\sum_{j=1}^{p}(U_j - \mu)_+^2 + (-U_j - \mu)_+^2 \le 4pe^{-\mu^2/2} = 4s/(p/s)^\xi.$$

By Markov's inequality, for this value of $\mu$ and $\xi > 0$,

$$\mathbb{P}\left(\frac{1}{s}\sum_{j=1}^{p}(|U_j| - \mu)_+^2 \le 2\xi^2\log(p/s)\right) \ge 1 - \frac{4}{2\xi^2\log(p/s)(p/s)^\xi}. \quad (40)$$

By the triangle inequality, on this event, we also have

$$\max_{A \subset \{1,...,p\}:|A|=s} \left( \frac{1}{s} \sum_{j \in A} U_j^2 \right)^{1/2} \leq \mu + \xi\sqrt{2\log(p/s)} = (1 + 2\xi)\sqrt{2\log(p/s)}.$$

Furthermore, if $\hat{A}$ denotes the subset achieving the maximum in the left hand side above, any value $U_j^2$ with $j \notin \hat{A}$ is smaller than the average of the values in $\hat{A}$ and

$$\max_{j \notin \hat{A}} |U_j| \leq \max_{A \subset \{1,...,p\}:|A|=s} \left( \frac{1}{s} \sum_{j \in A} U_j^2 \right)^{1/2} \leq (1 + 2\xi)\sqrt{2\log(p/s)}. \tag{41}$$

In linear regression with the squared loss, the vector $Z$ is $L\sigma^*$-subgaussian in the sense that $\mathbb{E}\exp(u^T Z) \leq \exp(L^2(\sigma^*)^2\|u\|^2/2)$ holds. Hence, since $\Sigma$ satisfies the normalization (**(N1)**), the random vector $U = \Sigma^{1/2}Z/(L\sigma^*)$ satisfies for all $j = 1,...,p$ that $\mathbb{E}[\exp(tU_j)] \leq e^{-t^2/2}$ and the bound (41) holds on an event of probability at least equal to the right hand side of (40).

For any $\xi > 0$, if $\lambda = L\sigma^*(1 + 3\xi)\sqrt{2\log(p/s)/n}$ then any $b \in \hat{T}$ where $\hat{T}$ is defined in (37) satisfies

$$0 \leq Z^T\Sigma^{1/2}(b-\beta^*) - \lambda\sqrt{n}(\|b\|_1 - \|\beta^*\|_1) = L\sigma^*\left(U^T(b - \beta^*) - \lambda\sqrt{n}(L\sigma^*)^{-1}(\|b\|_1 - \|\beta^*\|_1)\right).$$

This implies, by replacing $\lambda$ by its value and using the Cauchy-Schwarz inequality on the support of $\beta^*$, that

$$0 \leq U^T\delta + (1 + 3\xi)\sqrt{2\log(p/s)}(\sqrt{s}\|\delta\|_2 - \|b_{S^c}\|_1)$$

where $\delta = b - \beta^*$ and $S = \text{supp}(\beta^*)$. Then each component of $U$ satisfies $\mathbb{E}[\exp(tU_j)] \leq e^{-t^2/2}$ as explained above, the bound (40) applies. Hereafter, assume that event (41) holds. On event (41), $\sum_{j \in S} U_j \delta_j \leq (1 + 2\xi)\sqrt{2\log(p/s)}\|\delta\|_2$ by the Cauchy-Schwarz inequality. If $\hat{A} \subset \{1,...,p\}$ contains the indices of the $s$ largest coefficients of $U$ in absolute value, then $\sum_{j \in \hat{A}} U_j \delta_j \leq (1 + 2\xi)\sqrt{2\log(p/s)}\|\delta\|_2$ again by the Cauchy-Schwarz inequality. Finally, $\sum_{j \notin S \cup \hat{A}} U_j \delta_j \leq (1 + 2\xi)\sqrt{2\log(p/s)}\|\delta_{S^c}\|_1$ because $\max_{j \notin S \cup \hat{A}} |U_j|$ is bounded from above as in (41). Combining the above inequalities, on the event (41) we have

$$0 \leq \sqrt{s}\|\delta\|_2(2 + 5\xi)\sqrt{2\log(p/s)} - \xi\sqrt{2\log(p/s)}\|\delta_{S^c}\|_1.$$

This implies that $\|\delta_{S^c}\|_1 \leq \sqrt{s}\|\delta\|_2(2\xi^{-1} + 5)$ and $\|\delta\|_1 \leq \sqrt{s}\|\delta\|_2(6 + 2\xi^{-1})$.

The proof in logistic regression is the same up to a different scaling due to $\tilde{Z}$ from Lemma A.1(ii) being $L/2$-subgaussian, while in linear regression with the squared loss we had $Z$ being $L\sigma^*$-subgaussian. □

## B   Group-Lasso

**Lemma 3.5.** *Consider the linear model* (10) *and assume that* $\max_{k=1,...,M} \|\Sigma_{G_k,G_k}\|_{op} \leq 1$ *and that each group has the same size* $|G_k| = d = p/M$. *Let* $\xi > 0$ *and set* $\lambda = L\sigma^*(1 + \xi)[\sqrt{d} + (1 + 2\xi)\sqrt{2\log(M/s)}]$ *where* $(\sigma^*)^2 = (\sum_{i=1}^n \varepsilon_i^2)/n$ *and* $s$ *is the number of groups with* $\beta_{G_k}^* \neq 0$. *Then*

$$\mathbb{P}\left(\{\hat{\beta} - \beta^*, \eta - \beta^*\} \subset T\right) \geq 1 - 2/\left(2\xi^2\log(M/s)(M/s)^\xi\right). \tag{30}$$

*for* $T = \{\delta \in \mathbb{R}^p : \sum_{k=1}^M \|\delta_{G_k}\| \leq \sqrt{s}\|\delta\|_2(2 + 3\xi^{-1})\}$. *If instead the logistic regression model and assumptions of Proposition 2.2 are fulfilled and* $\lambda$ *is as above with* $\sigma^* = 1/2$, *then* (30) *also holds.*

*Proof of Lemma 3.5.* Define $U = \Sigma^{1/2}Z/(L\sigma^*)$ where $Z$ is as in Lemma A.1(i). Then $\mathbb{E}[\exp(v^T Z/(L\sigma^*))] \leq \exp(\|v\|^2/2)$ by the properties of $Z$ stated in Lemma A.1(i). We wish to study the restriction $U_{G_k}$ of $U$ to group $G_k$. Let $M_k$ be the matrix with $|G_k|$ rows and $p$ columns

made of the rows of $\Sigma^{1/2}$ indexed in $G_k$. Then $U_{G_k} = M_k Z/(L\sigma^*)$ and by applying the concentration inequality in [18] to the subgaussian vector $Z/(L\sigma^*)$ and the matrix $M_k^T M_k \in \mathbb{R}^{p\times p}$,

$$\|U_{G_k}\|_2^2 \le \mathsf{trace}(M_k^T M_k) + 2\sqrt{x}\mathsf{trace}(M_k^T M_k M_k^T M_k)^{1/2} + 2x\|M_k^T M_k\|_{op}.$$

with probability at least $1 - e^{-x}$. By properties of the trace, $\mathsf{trace}(M_k^T M_k) = \mathsf{trace}(M_k M_k^T) = \mathsf{trace}(\Sigma_{G_k,G_k})$. Similarly for the second term, $\mathsf{trace}(M_k^T M_k M_k^T M_k)^{1/2} = \|\Sigma_{G_k,G_k}\|_F$. Finally, $\|M_k^T M_k\|_{op} = \|M_k M_k^T\|_{op} = \|\Sigma_{G_k,G_k}\|_{op}$ so that the previous display reads

$$\|U_{G_k}\|_2^2 \le \mathsf{trace}(\Sigma_{G_k,G_k}) + 2\sqrt{x}\|\Sigma_{G_k,G_k}\|_F + 2x\|\Sigma_{G_k,G_k}\|_{op} \tag{42}$$

$$\le d + 2\sqrt{xd} + 2x \le (\sqrt{d} + \sqrt{2x})^2 \tag{43}$$

where for the second inequality we used that $\mathsf{trace}(\Sigma_{G_k,G_k}) \le |G_k| = d$ and $\|\Sigma_{G_k,G_k}\|_F^2 \le \sqrt{|G_k|} = \sqrt{d}$ using the assumption $\|\Sigma_{G_k,G_j}\|_{op} \le 1$. Hence $W_k = (\|U_{G_k}\|_2 - \sqrt{d})_+$ is 1-subgaussian for every group $k = 1, ..., M$, in the sense that $\mathbb{P}(W_k > \sqrt{2x}) \le e^{-x}$. As previously for the lasso, $W_k$ is thus stochastically dominated by $\sqrt{2\tau}$ where $\tau$ is an exponential random variable with parameter 1, and

$$\mathbb{E}[(\|U_{G_k}\|_2 - \sqrt{d} - \mu)_+^2] \le \mathbb{E}[(\sqrt{2\tau} - \mu)_+^2] \le 2\int_0^\infty (t - \mu^2/2)_+^2 e^{-t} dt = 2e^{-\mu^2/2}. \tag{44}$$

Define $W \ge 0$ by $W^2 = \sum_{k=1}^M (\|U_{G_k}\|_2 - \sqrt{d} - \mu)_+^2$. We have thus proved that $\mathbb{E}[W^2] \le 2Me^{-\mu^2/2}$ and for $\mu = (1+\xi)\sqrt{2\log(M/s)}$ we obtain $\mathbb{E}[W^2] \le 2s/(M/s)^\xi$, and by Markov's inequality

$$\mathbb{P}\left(\tfrac{1}{s}W^2 \le \xi^2 2\log(M/s)\right) \ge 1 - 2/(\xi^2 \log(M/s)(M/s)^\xi). \tag{45}$$

Furthermore, on the above event (45), by the triangle inequality we have

$$\max_{A\subset\{1,...,M\}:|A|=s} \left(\frac{1}{s}\sum_{k\in A}\|U_{G_k}\|_2^2\right)^{1/2} \le \sqrt{d} + \mu + \xi\sqrt{2\log(M/s)} = \sqrt{d} + (1+2\xi)\sqrt{2\log(M/s)} \triangleq \lambda_0.$$

Let $\lambda_0$ be defined as the right hand side of the previous display and notice that $\sqrt{n}\lambda/(L\sigma^*) = (1+\xi)\lambda_0$, so that if $\hat{A}$ is the subset of $[M]$ with the indices $k$ with largest $\|U_{G_k}\|$ (i.e., a subset attaining the maximum in the previous display), we have proved that

$$\max_{k\in\hat{A}^c}\|(\Sigma^{1/2}Z)_{G_k}\| \le \left(\frac{1}{s}\sum_{k\in\hat{A}}\|(\Sigma^{1/2}Z)_{G_k}\|_2^2\right)^{1/2} \le (1+\xi)^{-1}\sqrt{n}\lambda. \tag{46}$$

By Lemma A.1(i), and the Cauchy-Schwarz inequality on the groups in $S$,

$$0 \le Z^T\Sigma^{1/2}\delta + \sqrt{n}\lambda\sum_{k=1}^M(\|\beta_{G_k}^*\| - \|b_{G_k}\|) \le Z^T\Sigma^{1/2}\delta + \sqrt{n}\lambda\left(\sqrt{s}\|\delta\| - \sum_{k\notin S}\|b_{G_k}\|\right)$$

$$= L\sigma^*\left[\delta^T U + (1+\xi)\lambda_0\left(\sqrt{s}\|\delta\| - \sum_{k\notin S}\|b_{G_k}\|\right)\right],$$

where $\delta = (b - \beta^*)$ and $U = \Sigma^{1/2}Z/(L\sigma^*)$. We now bound $U^T\delta$ which appears on the previous display. On the above event, (45) we have $\sum_{k\in S}\delta_{G_k}{}^T U_{G_k} \le (\sum_{k\in S}\|U_{G_k}\|^2)^{1/2}\|\delta\| \le \sqrt{s}(\sqrt{d} + (1+2\xi)\sqrt{2\log(M/s)})\|\delta\| = \sqrt{s}\lambda_0\|\delta\|$. Similarly, if $\hat{A}$ contains the indices of the $s$ groups with the largest $\|U_{G_k}\|$ then on the above event (45), $\sum_{k\in\hat{A}}\delta_{G_k}{}^T U_{G_k} \le (\sum_{k\in\hat{A}}\|U_{G_k}\|^2)^{1/2}\|\delta\| \le \sqrt{s}(\sqrt{d} + (1+2\xi)\sqrt{2\log(M/s)})\|\delta\| = \sqrt{s}\lambda_0\|\delta\|$. For any group $G_k$ with $k \notin S\cup\hat{A}$, we have $\|U_{G_k}\| \le \lambda_0$. Combining these bounds with the fact that $\lambda = L\sigma^*(1+\xi)\lambda_0$, we have established that on event (45),

$$0 \le (3+\xi)\sqrt{s}\lambda_0\|\delta\| - \xi\lambda_0\sum_{k\notin S}\|\delta_{G_k}\|, \qquad \sum_{k\notin S}\|\delta_{G_k}\| \le \sqrt{s}(1 + 3/\xi)\|\delta\|. \tag{47}$$

On the groups indexed in $S$, by the Cauchy-Schwarz inequality we have $\sum_{k\in S}\|\delta_{G_k}\| \le \sqrt{s}\|\delta\|$. Hence $\delta = b - \beta^*$ belongs to the cone defined in the statement of the Lemma. $\qquad\square$

**Lemma 3.6.** *Assume that $\max_{k=1,\ldots,M} \|\Sigma_{G_k,G_k}\|_{op} \leq 1$ and that each group has the same size $|G_k| = d = p/M$. The set $T$ defined in the previous lemma satisfies $\gamma(T,\Sigma) \lesssim C(\xi)\phi(T)^{-1}\sqrt{sd + s\log(M/s)}$ for some constant $C(\xi)$ that depends only on $\xi$.*

*Proof of Lemma 3.6.* By definition of the restricted eigenvalue $\phi(T)$, for any $u \in T$ with $\|\Sigma^{1/2}u\| = 1$ we have $\|u\| \leq \phi(T)^{-1}$. Let $g \sim N(0, I_p)$; we wish to bound the expectation $\mathbb{E}\sup_{u\in T:\|\Sigma^{1/2}u\|=1} |g^T\Sigma^{1/2}u|$. Let $U = \Sigma^{1/2}g$. Let $\mu = \sqrt{2\log(M/s)}$. We have for any $u \in T$ with $\|\Sigma^{1/2}u\| = 1$,

$$u^T U \leq \sum_{k=1}^{M} \|u_{G_k}\|\|U_{G_k}\| = \sum_{k=1}^{M} \|u_{G_k}\|(\|U_{G_k}\| - \mu - \sqrt{d}) + (\mu + \sqrt{d})\sum_{k=1}^{M} \|u_{G_k}\|.$$

For the second term, since $u \in T$, inequality $\sum_{k=1}^{M} \|u_{G_k}\| \leq \sqrt{s}(2 + 3/\xi)\|u\|$ hence the second term is bounded from above as follows, $(\mu + \sqrt{d})\sum_{k=1}^{M} \|u_{G_k}\| \leq (\mu + \sqrt{d})\sqrt{s}(2 + 3/\xi)\phi(T)^{-1}$.

It remains to bound the first term. By the Cauchy-Schwarz inequality,

$$\sum_{k=1}^{M} \|u_{G_k}\|(\|U_{G_k}\| - \mu - \sqrt{d}) \leq \|u\| \left(\sum_{k=1}^{M}(\|U_{G_k}\| - \mu - \sqrt{d})_+^2\right)^{1/2}.$$

Finally, $\|u\| \leq \phi(T)^{-1}$ and the expectation bound (44) show that the previous display is bounded from above by $\phi(T)^{-1}(M2e^{-\mu^2/2})^{1/2} = \phi(T)^{-1}\sqrt{2s}$.

$\square$

# C    Proofs of Section 4

**Theorem 4.1.** *[Exact Risk Identity] Consider the linear model* (10) *and the regularized problem* (1) *with an arbitrary proper convex function $h(\cdot)$. Assume that $X_1, \ldots, X_n$ are iid $N(0, I_p)$ independent of $\varepsilon_1, \ldots, \varepsilon_n$ and set $\sigma^* = (\frac{1}{n}\sum_{i=1}^n \varepsilon_i^2)^{1/2}$. Then with probability at least $1 - 2\exp(-t^2/2)$,*

$$\left| \|\hat{\beta} - \beta^*\| - \mathbb{E}_Z\left[\|\beta^* - \mathsf{prox}_h(\beta^* + n^{-1/2}\sigma^* Z)\|^2\right]^{1/2} \right| \leq \frac{\sigma^*(t+1)}{n^{1/2}} + \|\hat{\beta} - \eta\| \qquad (31)$$

*where $Z = \frac{1}{n^{1/2}\sigma^*}\sum_{i=1}^n \varepsilon_i X_i \sim N(0, I_p)$ and $\mathbb{E}_Z$ denotes the expectation with respect $Z$.*

*Proof of Theorem 4.1.* Since $(X_i)_{i=1}^n$ and $(\varepsilon_i)_{i=1}^n$ are independent, we will condition throughout on $\varepsilon_1, \ldots, \varepsilon_n$. The proximal operator is Lipschitz for any convex function $h$, in the sense that

$$\|\mathsf{prox}_h(\beta^* + z) - \mathsf{prox}_h(\beta^* + z')\| \leq \|z - z'\| \quad \text{for all} \quad z, z' \in \mathbb{R}^p.$$

see Definition 2.3 and the following discussion in [27]. If $\Sigma = I_p$, the first order expansion $\eta$ in (5) is given by $\eta = \mathsf{prox}_h\left(\beta^* + n^{-1}\sum_{i=1}^n X_i\varepsilon_i\right)$. This means $\eta = \mathsf{prox}_h(\beta^* + n^{-1/2}\sigma^* Z)$ for $Z = (\sum_{i=1}^n \varepsilon_i^2)^{-1/2}\sum_{i=1}^n \varepsilon_i X_i$ and $Z \sim N(0, I_p)$ if $X_1, \ldots, X_n$ are iid $N(0, I_p)$ independent of $\varepsilon_1, \ldots, \varepsilon_n$. Note that in this case, $Z$ is independent of $\sigma^*$. Thus $\eta$ is a $n^{-1/2}\sigma^*$-Lipschitz function of $Z$, and by the triangle inequality $\|\eta - \beta^*\|$ is also a $n^{-1/2}\sigma^*$-Lipschitz function of $Z$. By the Gaussian concentration inequality [12, Theorem 5.6], with probability $1 - 2\exp(-t^2/2)$ we have

$$\left|\|\eta - \beta^*\| - \mathbb{E}_Z\left[\|\eta - \beta^*\|\right]\right| \leq n^{-1/2}\sigma^* t.$$

The Gaussian Poincaré inequality [12, Theorem 3.20] implies that $(\text{Var}(\|\eta - \beta^*\|))^{1/2}$ is bounded by the Lipschitz constant and hence $|\mathbb{E}_Z\left[\|\eta - \beta^*\|\right] - (\mathbb{E}_Z[\|\eta - \beta^*\|^2])^{1/2}| \leq n^{-1/2}\sigma^*$. Combining these inequalities above, we get with probability $1 - 2\exp(-t^2/2)$

$$\left|\|\eta - \beta^*\| - (\mathbb{E}[\|\beta^* - \mathsf{prox}_h(\beta^* + n^{-1/2}\sigma^* Z)\|^2])^{1/2}\right| \leq n^{-1/2}\sigma^*(t+1).$$

Therefore, by triangle inequality, on the same event we have

$$\left|\|\hat{\beta} - \beta^*\| - (\mathbb{E}[\|\beta^* - \mathsf{prox}_h(\beta^* + n^{-1/2}\sigma^* Z)\|^2])^{1/2}\right| \leq n^{-1/2}\sigma^*(t+1) + \|\eta - \hat{\beta}\|$$

which completes the proof. $\square$

# D Proofs of Section 5

**Proposition 5.1.** *Assume that $X_1, ..., X_n$ are iid $N(0, \Sigma)$ and is independent of $\varepsilon = (\varepsilon_1, ..., \varepsilon_n) \sim N(0, I_n)$. Assume that for some cone $T$ and $r_n = \gamma(T, \Sigma)/\sqrt{n}$ we have*

$$\mathbb{P}(\|\Sigma^{1/2}(\hat{\beta} - \beta^*)\| + \|\Sigma^{1/2}(\eta - \beta^*)\| \leq C_{19}r_n, \{\eta - \hat{\beta}, \eta - \beta^*, \hat{\beta} - \beta^*\} \subset T) \geq 1 - \alpha. \quad (33)$$

*Then for some $T_n$ with the t-distribution with $n$ degrees-of-freedom, with probability $1 - \alpha - 4e^{-nr_n^2/2}$,*

$$\sqrt{n}(\hat{\theta} - a^T\beta^*) - T_n = O_p((1 + r_n))\|\Sigma^{1/2}(\eta - \beta^*)\| + O_p(\sqrt{n}r_n)\|\Sigma^{1/2}(\eta - \hat{\beta})\|, \quad (34)$$

$$= O_p(r_n(1 + r_n)) + O_p(\sqrt{n}r_n^3). \quad (35)$$

*Proof of Proposition 5.1.* Some of the argument below is borrowed from [4, Section 6]. Let $T_n = \sqrt{n}\|z_a\|^{-2}z_a^T\varepsilon$, and let $Q_a = I_p - \Sigma^{-1}aa^T$; notice that $T_n$ has the t-distribution with $n$ degrees-of-freedom. Also note that $\Sigma^{1/2}Q_a\Sigma^{-1/2} = I - (\Sigma^{-1/2}a)(\Sigma^{-1/2}a)^\top$ and $z_a = X\Sigma^{-1}a = X\Sigma^{-1/2}\Sigma^{-1/2}a$; this implies that $XQ_a$ is independent of $z_a$ because $(z_a, XQ_a)$ are jointly normal and uncorrelated. By definition of $\hat{\theta}$, simple algebra yields that

$$\sqrt{n}(\hat{\theta} - a^T\beta^*) - T_n = -\sqrt{n}\|z_a\|^{-2}z_a^T XQ_a(\hat{\beta} - \beta^*) \quad (48)$$

$$= -\sqrt{n}\|z_a\|^{-2}z_a^T XQ_a(\eta - \beta^*) + \sqrt{n}\|z_a\|^{-2}z_a^T XQ_a(\eta - \hat{\beta}). \quad (49)$$

Note that $\eta$ only depends on $X$ through $\varepsilon^T X$ hence $\eta$ is independent of $(P_\varepsilon^\perp X, P_\varepsilon^\perp z_a)$ where $P_\varepsilon^\perp = I_n - \|\varepsilon\|^{-2}\varepsilon\varepsilon^T$ is an orthogonal projection; set also $P_\varepsilon = \|\varepsilon\|^{-2}\varepsilon\varepsilon^T$ for the complementary projection. We further split the first term above, so that $\sqrt{n}(\hat{\theta} - a^T\beta^*) - T_n$ is equal to

$$\sqrt{n}\|z_a\|^{-2}\left(-[z_a^T P_\varepsilon^\perp XQ_a(\eta - \beta^*)] - [z_a^T P_\varepsilon XQ_a(\eta - \beta^*)] + [z_a^T XQ_a(\eta - \hat{\beta})]\right)$$

Hereafter, assume that the event $\{\eta - \beta^*, \eta - \hat{\beta}\} \subset T$ holds (this holds with probability at least $1 - \alpha$ by assumption). For the first bracket inside the large parenthesis, $P_\varepsilon^\perp z_a$ is independent of $(\eta, XQ_a)$ conditionally on $\varepsilon$, so that $z_a^T P_\varepsilon^\perp XQ_a(\eta - \beta^*) = O_p(1)\|XQ_a(\eta - \beta^*)\|$. For the second bracket, since $P_\varepsilon$ is rank 1, $\|P_\varepsilon z_a\| = O_p(1)$ and the second bracket is also $O_p(1)\|XQ_a(\eta - \beta^*)\|$ by the Cauchy-Schwarz inequality. Since $\eta - \beta^* \in T$ and $r_n = \gamma(T, \Sigma)/\sqrt{n}$ we have $X = XQ_a + z_a a^T$ so that by the triangle inequality,

$$\|XQ_a(\eta - \beta^*)\| \leq \|X(\eta - \beta^*)\| + \|z_a\||a^T(\eta - \beta^*)| \leq \|X(\eta - \beta^*)\| + \|z_a\|\|\Sigma^{1/2}(\eta - \beta^*)\|.$$

By an application of [28], $\sup_{u \in T: \|\Sigma^{1/2}u\|=1} |\|Xu\| - \sqrt{n}\|\Sigma^{1/2}u\|| \leq C_{20}(\gamma(T, \Sigma) + t)$ with probability at least $1 - 2e^{-t^2/2}$, since $X\Sigma^{-1/2}$ has iid $N(0, 1)$ entries. We take $t = \gamma(T, \Sigma) = r_n\sqrt{n}$. Since $\|z_a\| = O_p(\sqrt{n})$, the first two brackets above are $O_p(\sqrt{n}(1 + r_n))\|\Sigma^{1/2}(\eta - \beta^*)\|$.

We now focus on the third bracket. Since $\eta - \hat{\beta} \in T$,

$$z_a^T XQ_a(\eta - \hat{\beta}) \leq \|\Sigma^{1/2}(\eta - \hat{\beta})\|\|z_a\| \sup_{u \in T: \|\Sigma^{1/2}u\|=1} \|z_a\|^{-1}z_a^T XQ_a\Sigma^{-1/2}\Sigma^{1/2}u. \quad (50)$$

Since $z_a$ and $XQ_a$ are independent, conditionally on $z_a$, the random vector $\|z_a\|^{-1}z_a^T XQ_a\Sigma^{-1/2}$ is normal with covariance matrix $\Sigma^{1/2}Q_a\Sigma^{-1/2}$. Since $\Sigma^{1/2}Q_a\Sigma^{-1/2}$ is a projection matrix in $\mathbb{R}^p$, $\Sigma^{1/2}Q_a\Sigma^{-1/2} \preceq I_p$ in the sense of positive semi-definite matrices. By the Sudakov-Fernique's inequality [43, Theorem 7.2.11], conditionally on $z_a$ we have

$$\mathbb{E}\left[\sup_{u \in T: \|\Sigma^{1/2}u\|=1} \|z_a\|^{-1}z_a^T XQ_a\Sigma^{-1/2}\Sigma^{1/2}u \mid z_a\right] \leq \mathbb{E}\left[\sup_{u \in T: \|\Sigma^{1/2}u\|=1} g^T\Sigma^{1/2}u\right] = \gamma(T, \Sigma)$$

for some $g \sim N(0, I_p)$. Furthermore, by Gaussian concentration [12, Theorem 5.8] again conditionally on $z_a$ we have $\sup_{u \in T: \|\Sigma^{1/2}u\|=1} \|z_a\|^{-1}z_a^T XQ_a\Sigma^{-1/2}\Sigma^{1/2}u \leq \gamma(T, \Sigma) + t$ with probability at least $1 - e^{-t^2/2}$. Taking $t = \gamma(T, \Sigma) = r_n\sqrt{n}$, we obtain that the right hand side of (50) is bounded from above on an event of probability at least $1 - e^{-t^2/2}$ by $\|\Sigma^{1/2}(\eta - \hat{\beta})\|\|z_a\|\sqrt{n}r_n$. Given that $\|z_a\|/\sqrt{n} = O_p(1)$ and $\sqrt{n}/\|z_a\| = O_P(1)$, the proof is complete. $\qquad\square$

# E  Proofs of Results in 6

**Theorem 6.1.** *Define $\hat{K} := n^{-1} \sum_{i=1}^{n} \ell''(Y_i, X_i^\top \beta^*) X_i X_i^\top$. Under assumption (A1), we have*

*(i) If $\{\hat{\beta} - \beta^*, \eta - \beta^*\} \subseteq T$ then $\|\hat{\beta} - \eta\|_K \lesssim Q_{n,1}^{1/2} \mathcal{E} + B^{1/2} Z_n^{1/2} \mathcal{E}^{3/2}$.*

*(ii) If $\{\hat{\beta} - \eta, \hat{\beta} - \beta^*, \eta - \beta^*\} \subseteq T$ then $\|\hat{\beta} - \eta\|_K \lesssim Q_{n,2} \mathcal{E} + B Z_n \mathcal{E}^2$,*

*where*

$$Q_{n,1} = \sup_{u \in T} \left| \frac{u^\top \hat{K} u}{\|u\|_K^2} - 1 \right|, \quad Q_{n,2} = \sup_{u,v \in T} \frac{|u^\top (\hat{K} - K) v|}{\|u\|_K \|v\|_K} \quad and \quad Z_n = \sup_{u \in T} \frac{1}{n} \sum_{i=1}^{n} \frac{|X_i^\top u|^3}{\|u\|_K^3}.$$

*Proof of Theorem 6.1.* By strong convexity of the objective function of $\eta$ with respect to the norm $\|\cdot\|_K$, or alternatively by application of for instance [6, Lemma 1] or or [7, Lemma A.2],

$$\frac{1}{2} \|K^{1/2}(\hat{\beta} - \eta)\|^2 \le \frac{1}{n} \sum_{i=1}^{n} \ell'(Y_i, X_i^\top \beta^*) X_i^\top (\hat{\beta} - \beta^*) + \frac{1}{2} \|K^{1/2}(\hat{\beta} - \beta^*)\|^2 + h(\hat{\beta}) \tag{51}$$

$$- \left[ \frac{1}{n} \sum_{i=1}^{n} \ell'(Y_i, X_i^\top \beta^*) X_i^\top (\eta - \beta^*) + \frac{1}{2} \|K^{1/2}(\eta - \beta^*)\|^2 + h(\eta) \right].$$

From the definition (1) of $\hat{\beta}$, we get

$$0 \le \frac{1}{n} \sum_{i=1}^{n} \ell(Y_i, X_i^\top \eta) + h(\eta) - \frac{1}{n} \sum_{i=1}^{n} \ell(Y_i, X_i^\top \hat{\beta}) - h(\hat{\beta}). \tag{52}$$

Adding inequalities (51) and (52), the terms $h(\hat{\beta})$ and $h(\eta)$ cancel out and we obtain

$$\frac{1}{2} \|K^{1/2}(\hat{\beta} - \eta)\|^2$$

$$\le \frac{1}{n} \sum_{i=1}^{n} \ell(Y_i, X_i^\top \eta) - \frac{1}{n} \sum_{i=1}^{n} \ell'(Y_i, X_i^\top \beta^*) X_i^\top (\eta - \beta^*) - \frac{1}{2} \|K^{1/2}(\eta - \beta^*)\|^2$$

$$- \left[ \frac{1}{n} \sum_{i=1}^{n} \ell(Y_i, X_i^\top \hat{\beta}) - \frac{1}{n} \sum_{i=1}^{n} \ell'(Y_i, X_i^\top \beta^*) X_i^\top (\hat{\beta} - \beta^*) - \frac{1}{2} \|K^{1/2}(\hat{\beta} - \beta^*)\|^2 \right]. \tag{53}$$

The terms on the right hand side resemble the remainder terms from a second order Taylor series expansion except for $K$ instead of $\hat{K}$. Using the Taylor expansion above, we get for any $\beta \in \mathbb{R}^p$

$$\frac{1}{n} \sum_{i=1}^{n} \ell(Y_i, X_i^\top \beta) - \frac{1}{n} \sum_{i=1}^{n} \ell'(Y_i, X_i^\top \beta^*) X_i^\top (\beta - \beta^*) - \frac{1}{2} \|K^{1/2}(\beta - \beta^*)\|^2 \tag{54}$$

$$= \frac{1}{2n} \sum_{i=1}^{n} |X_i^\top (\beta - \beta^*)|^2 a_i(\beta) + \frac{1}{2}(\beta - \beta^*)^\top (\hat{K} - K)(\beta - \beta^*), \tag{55}$$

where

$$a_i(\beta) := \int_0^1 \{ \ell''(Y_i, X_i^\top \beta^* + t X_i^\top (\beta - \beta^*)) - \ell''(Y_i, X_i^\top \beta^*) \} dt. \tag{56}$$

(Note that for the squared loss, $\ell'' = 1$ is constant and $a_i(\cdot) = 0$ which leads to a mucher simple analysis; the reader only interested in squared loss may skip the analysis of $a_i(\cdot)$). Substituting this in (53), we get

$$\|K^{1/2}(\hat{\beta} - \eta)\|^2 \le (\eta - \beta^*)^\top (\hat{K} - K)(\eta - \beta^*) - (\hat{\beta} - \beta^*)^\top (\hat{K} - K)(\hat{\beta} - \beta^*) \tag{57}$$

$$+ \frac{1}{n} \sum_{i=1}^{n} |X_i^\top (\eta - \beta^*)|^2 a_i(\eta) - \frac{1}{n} \sum_{i=1}^{n} |X_i^\top (\hat{\beta} - \beta^*)|^2 a_i(\hat{\beta}). \tag{58}$$

From the Lipschitz condition on $\ell''(\cdot, \cdot)$, we get $|a_i(\beta)| \leq B|X_i^\top(\beta - \beta^*)|$, and hence part 1 of the result follows.

In part 1, we did not use any information about $\hat{\beta} - \eta$. For part 2, we will control the right hand side "quadratic forms" in (57) in a more refined way. By simple algebra and the definition of $Q_{n,2}(\cdot)$,

$$
\begin{aligned}
(\eta - \beta^*)^\top &(\hat{K} - K)(\eta - \beta^*) - (\hat{\beta} - \beta^*)^\top(\hat{K} - K)(\hat{\beta} - \beta^*) \\
&= ((\eta - \beta^*) - (\hat{\beta} - \beta^*))^\top(\hat{K} - K)((\eta - \beta^*) + (\hat{\beta} - \beta^*)) \\
&= (\eta - \hat{\beta})^\top(\hat{K} - K)(\eta - \beta^*) \\
&\quad + (\eta - \hat{\beta})^\top(\hat{K} - K)(\hat{\beta} - \beta^*) \\
&\leq Q_{n,2}(T)\|K^{1/2}(\eta - \hat{\beta})\| \left[\|K^{1/2}(\hat{\beta} - \beta^*)\| + \|K^{1/2}(\eta - \beta^*)\|\right].
\end{aligned}
\tag{59}
$$

This completes the control of first difference in (57). For the second difference in (58), observe that

$$
|a_i(\beta)| \leq B|X_i^\top(\beta - \beta^*)| \quad \text{and} \quad |a_i(\beta) - a_i(\alpha)| \leq B|X_i^\top(\beta - \alpha)|,
$$

and hence

$$
\begin{aligned}
\frac{1}{n}\sum_{i=1}^n &|X_i^\top(\eta - \beta^*)|^2 a_i(\eta) - \frac{1}{n}\sum_{i=1}^n |X_i^\top(\hat{\beta} - \beta^*)|^2 a_i(\hat{\beta}) \\
&= \frac{1}{n}\sum_{i=1}^n a_i(\eta)\left[|X_i^\top(\eta - \beta^*)|^2 - |X_i^\top(\hat{\beta} - \beta^*)|^2\right] + \frac{1}{n}\sum_{i=1}^n |X_i^\top(\hat{\beta} - \beta^*)|^2 \left[a_i(\eta) - a_i(\hat{\beta})\right] \\
&\leq \frac{B}{n}\sum_{i=1}^n |X_i^\top(\eta - \beta^*)| \times |X_i^\top((\eta - \beta^*) - (\hat{\beta} - \beta^*))| \times |X_i^\top((\eta - \beta^*) + (\hat{\beta} - \beta^*))| \\
&\quad + \frac{B}{n}\sum_{i=1}^n |X_i^\top(\hat{\beta} - \beta^*)|^2 \times |X_i^\top(\eta - \hat{\beta})| \\
&\leq \frac{B}{n}\sum_{i=1}^n |X_i^\top(\eta - \hat{\beta})| \times \left[|X_i^\top(\eta - \beta^*)|^2 + |X_i^\top(\hat{\beta} - \beta^*)|^2\right] \\
&\quad + \frac{B}{n}\sum_{i=1}^n |X_i^\top(\eta - \hat{\beta})| \times |X_i^\top(\eta - \beta^*)| \times |X_i^\top(\hat{\beta} - \beta^*)|.
\end{aligned}
$$

The right hand side is trivially bounded by

$$
3B\|\eta - \hat{\beta}\|_K \left[\|(\eta - \beta^*)\|_K^2 + \|(\hat{\beta} - \beta^*)\|_K^2\right] \sup_{u,v,w \in \tilde{T}} \frac{1}{n}\sum_{i=1}^n \frac{|X_i^\top u|}{\|K^{1/2}u\|} \times \frac{|X_i^\top v|}{\|K^{1/2}v\|} \times \frac{|X_i^\top w|}{\|K^{1/2}w\|}.
$$

Using $3abc \leq a^3 + b^3 + c^3$ for any positive $\{a, b, c\}$, the previous display is bounded from above by

$$
B\|K^{1/2}(\eta - \hat{\beta})\| \left[\|K^{1/2}(\eta - \beta^*)\|^2 + \|K^{1/2}(\hat{\beta} - \beta^*)\|^2\right] Z_n(T).
$$

Substituting these bounds in (58), we get the result. $\qquad\square$

**Proposition 6.2.** *[Control of $Q_{n,1}, Q_{n,2}$ and $Z_n$] Under assumptions (A1) and (A2), we have*
*(i) With probability $1 - 2\exp(-C_{21}t^2\gamma^2(T, \Sigma))$,*

$$
\max\{Q_{n,1}(T), Q_{n,2}(T)\} \leq C_{22}B_2B_3L^2\left(tn^{-1/2}\gamma(T, \Sigma) + t^2 n^{-1}\gamma^2(T, \Sigma)\right).
$$

*(ii) With probability $1 - 2\exp(-C_{23}t\log n)$,* $\quad Z_n(T) \leq C_{24}B_3^{3/2}L^3\left(1 + n^{-1}\gamma^3(T, \Sigma)\right)t^3.$

*Proof of Proposition 6.2.* Define the function classes $F$ and $H$ as

$$
\begin{aligned}
F &:= \left\{(x, y) \mapsto x^\top u/\|u\|_K : u \in T\right\}, \\
H &:= \left\{(x, y) \mapsto \ell''(y, x^\top\beta^*)x^\top u/\|u\|_K : u \in T\right\}.
\end{aligned}
$$

It is then clear that

$$\max\{Q_{n,1}(T), Q_{n,2}(T)\} \leq \sup_{f \in F, h \in H} \left| \frac{1}{n} \sum_{i=1}^{n} \{f(X_i, Y_i)h(X_i, Y_i) - \mathbb{E}[f(X_i)h(X_i)]\} \right|,$$

$$Z_n(T) = \sup_{f \in \mathcal{F}} \frac{1}{n} \sum_{i=1}^{n} |f(X_i)|^3.$$

We now apply Theorem 1.13 of [33] with $2^{s_0/2} = \gamma(T, \Sigma), q = 5$. Hence, we get for any $t \geq 8$ with probability at least $1 - 2\exp(-c_1 t^2 \gamma^2(T, \Sigma))$,

$$\sup_{f \in F, h \in H} \left| \frac{1}{n} \sum_{i=1}^{n} \{f(X_i, Y_i)h(X_i, Y_i) - \mathbb{E}[f(X_i, Y_i)h(X_i, Y_i)]\} \right| \qquad (60)$$

$$\leq \frac{c_2 t^2}{n} (\gamma(F) + 2^{s_0/2}\text{diam}(F))(\gamma(H) + 2^{s_0/2}\text{diam}(H))$$

$$+ \frac{c_2 t}{\sqrt{n}} (\text{diam}(F)(\gamma(H) + 2^{s_0/2}\text{diam}(H)) + \text{diam}(H)(\gamma(F) + 2^{s_0/2}\text{diam}(F))),$$

where

$$\gamma(F) := L\mathbb{E}\left[\sup_{u \in T} \frac{|g^\top \Sigma^{1/2} u|}{\|K^{1/2}u\|}\right] \leq L\sup_{u \in T} \frac{\|\Sigma^{1/2}u\|}{\|K^{1/2}u\|}\gamma(T, \Sigma) \leq B_3^{1/2}L\gamma(T, \Sigma),$$

$$\gamma(H) := B_2 L\mathbb{E}\left[\sup_{u \in T} \frac{|g^\top \Sigma^{1/2} u|}{\|K^{1/2}u\|}\right] \leq B_2 B_3^{1/2}L\gamma(T, \Sigma),$$

and

$$\text{diam}(F) := L\sup_{u \in T} \frac{\|\Sigma^{1/2}u\|}{\|K^{1/2}u\|} \leq B_3^{1/2}L,$$

$$\text{diam}(H) := B_2 L\sup_{u \in T} \frac{\|\Sigma^{1/2}u\|}{\|K^{1/2}u\|} \leq B_2 B_3^{1/2}L.$$

Substituting these quantities in (60), part 1 of the result follows. Alternatively, one could apply Theorem 1.12 of [33] if $\ell(\cdot, \cdot)$ is assumed to be convex in the second argument (implying $\ell''(y, x^\top \beta^*) \geq 0$).

To prove part 2, we apply Equation (3.9) of [32] with $|I| = n$. Hence, we have with probability $1 - 2\exp(-c_1 t \log n)$

$$Z_n(T) = \sup_{f \in F} \frac{1}{n} \sum_{i=1}^{n} |f(X_i)^3| \leq \frac{ct^3}{n} \left(\gamma(F) + n^{1/3}\text{diam}(F)\right)^3, \qquad (61)$$

where $c > 0$ is an absolute constant and

$$\gamma(F) := L\mathbb{E}\left[\sup_{u \in T} \frac{|g^\top \Sigma^{1/2} u|}{\|K^{1/2}u\|}\right] \leq B_3^{1/2}L\gamma(T, \Sigma),$$

and

$$\text{diam}(F) := L\sup_{u \in T} \frac{\|\Sigma^{1/2}u\|}{\|K^{1/2}u\|} \leq B_3^{1/2}L.$$

Hence part 2 follows. $\qquad\qquad\square$

### E.1 Verification of Assumption (A1) for Logistic Loss

**Proposition 2.2.** *Consider the logistic loss $\ell(y, u) = yu - \log(1 + e^u)$ for $y \in \{0, 1\}, u \in \mathbb{R}$. Assume that $(X_i, Y_i)_{i=1,\dots,n}$ are iid satisfying the logistic regression model*

$$\mathbb{P}(Y_i = 1|X_i) = 1 - \mathbb{P}(Y_i = 0|X_i) = 1/(1 + \exp(X_i^\top \beta^*)),$$

*for some $\beta^* \in \mathbb{R}^p$ with $\|\Sigma^{1/2}\beta^*\| \leq 1$.[2] Assume **(A2)** holds. Then (8) holds with $B = 1/(6\sqrt{3})$, $B_2 = 1$ and an absolute constant $B_3 > 0$.*

Lipschitzness and boundedness of $\ell''(y, u)$ for logistic loss is straightforward. These parts do not require $\|\Sigma^{1/2}\beta^*\| \leq 1$. In order to prove the third part, we prove the following general result for general loss function with a lower second-order curvature.

Define for any $t > 0$

$$\alpha(t) := \inf_{y \in \mathbb{R}} \inf_{|u| \leq t} \ell''(y, u).$$

Note that $\alpha(\cdot)$ is non-increasing. This is called the lower curvature function and appears in the works Loh [30, Section 3.2] and Koltchinskii [24, Section 3].

**Proposition E.1.** *Suppose $X_1, \ldots, X_n$ are iid $L$-subGaussian random vectors with covariance $\Sigma$. Then there exists absolute constants $c_1, c_2 > 0$ such that for any $u \in \mathbb{R}^p$, we have*

$$\frac{u^\top K_n u}{u^\top \Sigma u} \geq \sup_{\tau > 0} \alpha(\tau) \left\{ 1 - c_1 L (\mathbb{P}(|X_1^\top \beta^*| > \tau))^{1/2} \right\} \tag{62}$$

$$\geq \alpha(2L\sqrt{\log(c_2 L)}\|\Sigma^{1/2}\beta^*\|_2)/2. \tag{63}$$

*Proof.* Fix a number $\tau > 0$. It is clear that

$$u^\top K u = \mathbb{E}\left[\ell''(Y_1, X_1^\top \beta^*)(X_1^\top u)^2\right] \geq \alpha(\tau)\mathbb{E}\left[(X_1^\top u)^2 \mathbb{1}\{|X_1^\top \beta^*| \leq \tau\}\right]. \tag{64}$$

Observe now that for any $u \in \mathbb{R}^p$, we have

$$\begin{aligned}
0 &\leq \mathbb{E}[(X_1^\top u)^2] - \mathbb{E}[(X_1^\top u)^2 \mathbb{1}\{|X_1^\top \beta^*| \leq \tau\}] \\
&= \mathbb{E}[(X_1^\top u)^2 \mathbb{1}\{|X_1^\top \beta^*| > \tau\}] \\
&\leq (\mathbb{E}[(X_1^\top u)^4])^{1/2}(\mathbb{P}(|X_1^\top \beta^*| > \tau))^{1/2} \\
&\leq cL\|\Sigma^{1/2}u\|^2(\mathbb{P}(|X_1^\top \beta^*| > \tau))^{1/2},
\end{aligned}$$

for some absolute constant $c > 0$. This implies that

$$\mathbb{E}\left[(X_1^\top u)^2 \mathbb{1}\{|X_1^\top \beta^*| \leq \tau\}\right] \geq \|\Sigma^{1/2}u\|_2^2 \left\{ 1 - cL(\mathbb{P}(|X_1^\top \beta^*| > \tau))^{1/2} \right\}.$$

Combining this with (64), we get

$$\frac{u^\top K u}{u^\top \Sigma u} \geq \sup_{\tau > 0} \alpha(\tau) \left\{ 1 - cL(\mathbb{P}(|X_1^\top \beta^*| > \tau))^{1/2} \right\}.$$

This proves the first inequality. To prove the second inequality, take $\tau = \rho\|\Sigma^{1/2}\beta^*\|_2$ for some $\rho > 0$ (to be determined later). For this choice, we have from $L$-subGaussianity

$$\mathbb{P}\left(|X_1^\top \beta^*| \geq \tau\right) = \mathbb{P}(|X_1^\top \beta^*| \geq \rho\|\Sigma^{1/2}\beta^*\|_2) \leq 2\exp\left(-\frac{\rho^2}{2L^2}\right).$$

Hence, if $\rho = 2L(\log(\sqrt{8}cL))^{1/2}$ then

$$1 - cL(\mathbb{P}(|X_1^\top \beta^*| > \tau))^{1/2} \geq 1 - \sqrt{2}cL\exp\left(-\frac{\rho^2}{4L^2}\right) \tag{65}$$

$$= 1 - \exp\left(-\frac{\rho^2}{4L^2} + \log(\sqrt{2}cL)\right) \geq \frac{1}{2}. \tag{66}$$

Therefore, for any $u \in \mathbb{R}^p$,

$$\frac{u^\top K u}{u^\top \Sigma u} \geq \frac{1}{2}\alpha(2L(\log(\sqrt{8}cL))^{1/2}\|\Sigma^{1/2}\beta^*\|_2).$$

This completes the proof. $\qquad\square$

# F   Verification of Restricted Strong Convexity and Rates for Logistic Lasso

In the main paper, we proved/stated bounds for $\|\hat{\beta} - \beta^*\|_K$ and $\|\eta - \beta^*\|_K$ for squared loss with different penalties. These proofs can be extended to the case of logistic loss once restricted strong convexity (RSC) condition is verified; see Proposition 3.4 which is restated and proved below. Also, see [35] where the RSC was introduced. We present the following result that proves RSC for general cones $T$.

**Proposition F.1.** *Fix any cone $T \subset \mathbb{R}^p$. Assume (A1) and (A2) holds. If the loss function satisfies*

$$\sup_{|s-t| \leq u} \frac{\ell''(y, s)}{\ell''(y, t)} \leq \exp(3u), \tag{67}$$

*then for any $u \in T$ satisfying $\|u\|_K \leq 1$, we have*

$$
\frac{1}{n} \sum_{i=1}^n \ell(Y_i, X_i^\top \beta^* + X_i^\top u) - \frac{1}{n} \sum_{i=1}^n \ell(Y_i, X_i^\top \beta^*) - \frac{1}{n} \sum_{i=1}^n u^\top X_i \ell'(Y_i, X_i^\top \beta^*)
$$
$$
\geq \|u\|_K^2 \frac{B_2^{-1}[0.5 B_3^{-1/2} - \tilde{Q}_n(T)]}{\exp(6\sqrt{2} L (\log(4 B_3^{1/2} B_2 L^2))^{1/2})}, \tag{68}
$$

*for a random quantity $\tilde{Q}_n(T)$ which satisfies the following: there exists a universal constant $C > 0$ such that for any $t \geq 0$, we have with probability $1 - \exp(-t)$,*

$$Q_n(T) \leq C \left( \frac{B_2 L \gamma(T, \Sigma)}{\sqrt{n}} + \frac{B_2 \gamma^2(T, \Sigma)}{n} + \frac{t^{1/2} B_2 L^2}{\sqrt{n}} + \frac{t B_2 L^2}{n} \right).$$

Assumption (67) can be verified for logistic regression easily. Therefore, if $\gamma(T, \Sigma)/\sqrt{n} \to 0$ then for all $u \in T, \|u\|_K \leq 1$, we get

$$
\frac{1}{n} \sum_{i=1}^n \ell(Y_i, X_i^\top \beta^* + X_i^\top u) - \frac{1}{n} \sum_{i=1}^n \ell(Y_i, X_i^\top \beta^*) - \frac{1}{n} \sum_{i=1}^n u^\top X_i \ell'(Y_i, X_i^\top \beta^*) \geq \mathfrak{C} \|u\|_\Sigma^2,
$$

for a constant $\mathfrak{C}$ depending on $L, B, B_2, B_3$.

*Proof.* For notational convenience, let

$$f_n(\beta) := \frac{1}{n} \sum_{i=1}^n \ell(Y_i, X_i^\top \beta).$$

Define for $\beta^*$ and any $u \in T$,

$$\Delta_2 f_n(\beta^*, u) := f_n(\beta^* + u) - f_n(\beta^*) - u^\top \nabla f_n(\beta^*).$$

Note that

$$\Delta_2 f_n(\beta^*, u) = \frac{1}{n} \sum_{i=1}^n \ell''(Y_i, X_i^\top (\beta^* + su)) \langle X_i, u \rangle^2,$$

for some $s \in [0, 1]$. From the stability property (67) of $\ell''(\cdot, \cdot)$, we get that

$$\ell''(Y_i, X_i^\top (\beta^* + su)) \langle X_i, u \rangle^2 \geq \exp(-3|\langle u, X_i \rangle|) \ell''(Y_i, X_i^\top \beta^*) \langle X_i, u \rangle^2.$$

This implies that

$$
\Delta_2 f_n(\beta^*, u) = \frac{1}{n} \sum_{i=1}^n \ell''(Y_i, X_i^\top \beta^*) \exp(-3|\langle u, X_i \rangle|) \langle X_i, u \rangle^2
$$
$$
= \|u\|_\Sigma^2 \frac{1}{n} \sum_{i=1}^n \ell''(Y_i, X_i^\top \beta^*) \exp(-3|\langle u, X_i \rangle|) \left( \frac{\langle u, X_i \rangle}{\|u\|_\Sigma} \right)^2 \tag{69}
$$

Now define the function
$$\varphi_\tau(t) = \begin{cases} |t|, & \text{if } |t| \le \tau/2, \\ \tau - |t|, & \text{if } \tau/2 \le |t| \le \tau, \\ 0, & \text{otherwise.} \end{cases}$$

It is clear that
$$\frac{\langle X_i, u \rangle^2}{\|u\|_\Sigma^2} \ge \varphi_\tau^2 \left( \frac{\langle X_i, u \rangle}{\|u\|_\Sigma} \right).$$

Further $\varphi_\tau(\cdot)$ is a 1-Lipschitz function. Using these properties, we get

$$\begin{aligned}
\Delta_2 f_n(\beta^*, u) &\ge \|u\|_\Sigma^2 \frac{1}{n} \sum_{i=1}^n \ell''(Y_i, X_i^\top \beta^*) \exp(-3|\langle u, X_i \rangle|) \varphi_\tau^2 \left( \frac{\langle X_i, u \rangle}{\|u\|_\Sigma} \right) \\
&\ge \|u\|_\Sigma^2 \exp(-3\tau \|u\|_\Sigma) \frac{1}{n} \sum_{i=1}^n \ell''(Y_i, X_i^\top \beta^*) \varphi_\tau^2 \left( \frac{\langle X_i, u \rangle}{\|u\|_\Sigma} \right).
\end{aligned} \tag{70}$$

To complete the proof note that

$$\begin{aligned}
&\frac{1}{n} \sum_{i=1}^n \ell''(Y_i, X_i^\top \beta^*) \varphi_\tau^2 \left( \frac{\langle X_i, u \rangle}{\|u\|_\Sigma} \right) \\
&= \frac{1}{n} \sum_{i=1}^n \mathbb{E} \left[ \ell''(Y_i, X_i^\top \beta^*) \varphi_\tau^2 \left( \frac{\langle X_i, u \rangle}{\|u\|_\Sigma} \right) \right] \\
&\quad + \frac{1}{n} \sum_{i=1}^n \left\{ \ell''(Y_i, X_i^\top \beta^*) \varphi_\tau^2 \left( \frac{\langle X_i, u \rangle}{\|u\|_\Sigma} \right) - \mathbb{E} \left[ \ell''(Y_i, X_i^\top \beta^*) \varphi_\tau^2 \left( \frac{\langle X_i, u \rangle}{\|u\|_\Sigma} \right) \right] \right\}.
\end{aligned} \tag{71}$$

We now control the first term by noting that

$$\begin{aligned}
&\frac{1}{n} \sum_{i=1}^n \mathbb{E} \left[ \ell''(Y_i, X_i^\top \beta^*) \left\{ \left( \frac{\langle X_i, u \rangle}{\|u\|_\Sigma} \right)^2 - \varphi_\tau^2 \left( \frac{\langle X_i, u \rangle}{\|u\|_\Sigma} \right) \right\} \right] \\
&\le \frac{1}{n} \sum_{i=1}^n \mathbb{E} \left[ \ell''(Y_i, X_i^\top \beta^*) \left( \frac{\langle X_i, u \rangle}{\|u\|_\Sigma} \right)^2 \mathbb{1} \{ |\langle u, X_i \rangle| \ge \tau \|u\|_\Sigma/2 \} \right] \\
&\le B_2 \frac{1}{n} \sum_{i=1}^n \mathbb{E} \left[ \frac{\langle u, X_i \rangle^2}{\|u\|_\Sigma^2} \mathbb{1} \{ |\langle u, X_i \rangle| \ge \tau \|u\|_\Sigma/2 \} \right] \\
&\le B_2 \int_{\tau/2}^\infty 2t \exp\left( -\frac{t^2}{2L^2} \right) dt = 2 B_2 L^2 \exp\left( -\frac{\tau^2}{8L^2} \right),
\end{aligned} \tag{72}$$

using the boundedness of $\ell''$ and $L$-sub-Gaussianity of $X_i$. This implies that

$$\frac{1}{n} \sum_{i=1}^n \mathbb{E} \left[ \ell''(Y_i, X_i^\top \beta^*) \varphi_\tau^2 \left( \frac{\langle X_i, u \rangle}{\|u\|_\Sigma} \right) \right] \ge \frac{\|u\|_{K_n}}{\|u\|_\Sigma} - 2 B_2 L^2 \exp\left( -\frac{\tau^2}{8L^2} \right) dt \ge \frac{1}{2 B_3^{1/2}}, \quad (73)$$

for $\tau := 2\sqrt{2} L (\log(4 B_3^{1/2} B_2 L^2))^{1/2}$.

Combining this with the lower bound on $\Delta_2 f_n(\beta^*, u)$, we get for all $u \in T$,

$$\Delta_2 f_n(\beta^*, u) \ge \|u\|_\Sigma^2 \exp(-3\tau \|u\|_\Sigma) \left[ 0.5 B_3^{-1/2} - \tilde{Q}_n(T) \right],$$

where

$$\tilde{Q}_n(T) := \max_{u \in T} \left| \frac{1}{n} \sum_{i=1}^n \left\{ \ell''(Y_i, X_i^\top \beta^*) \varphi_\tau^2 \left( \frac{\langle X_i, u \rangle}{\|u\|_\Sigma} \right) - \mathbb{E} \left[ \ell''(Y_i, X_i^\top \beta^*) \varphi_\tau^2 \left( \frac{\langle X_i, u \rangle}{\|u\|_\Sigma} \right) \right] \right\} \right|.$$

Before bounding $\tilde{Q}_n(T)$, note from assumption **(A1)** that

$$\|u\|_K^2 \le B_2 \|u\|_\Sigma^2 \le B_2 B_3 \|u\|_K,$$

and hence for all $u \in T$

$$\Delta_2 f_n(\beta^*, u) \geq B_2^{-1} \|u\|_K^2 \exp(-3B_3^{1/2} \tau \|u\|_K) \left[ 0.5 B_3^{-1/2} - \tilde{Q}_n(T) \right], \tag{74}$$

We now bound $\tilde{Q}_n(T)$. Define the function class

$$F := \{(x, y) \mapsto (\ell''(y, x^\top \beta^*))^{1/2} \varphi_\tau(\langle u, x \rangle / \|u\|_\Sigma) : u \in T\}.$$

From this definition, it follows that

$$\tilde{Q}_n(T) = \sup_{f \in F} \left| \frac{1}{n} \sum_{i=1}^n \left\{ f^2(X_i, Y_i) - \mathbb{E}[f^2(X_i, Y_i)] \right\} \right|.$$

We now apply Theorem 5.5 of [16] to get with probability $1 - \exp(-t)$,

$$\tilde{Q}_n(T) \leq C \left( \frac{\gamma(F) \text{diam}(F)}{\sqrt{n}} + \frac{\gamma^2(F)}{n} + \frac{t^{1/2} \text{diam}^2(F)}{\sqrt{n}} + \frac{t \text{diam}^2(F)}{n} \right),$$

for some constant $C > 0$ where

$$\text{diam}(F) := \sup_{f \in F} \|f(X, Y)\|_{\psi_2} \leq \sup_{u \in T} \frac{\|(\ell''(Y, X^\top \beta^*))^{1/2} \varphi_\tau(\langle u, X \rangle / \|u\|_\Sigma)\|_{\psi_2}}{\|u\|_\Sigma}$$

$$\leq B_2^{1/2} \sup_{u \in T} \frac{\|\langle u, X \rangle\|_{\psi_2}}{\|u\|_\Sigma} \leq B_2^{1/2} L,$$

and

$$\gamma(F) := B_2^{1/2} \mathbb{E} \left[ \max_{u \in T} \frac{|\langle \Sigma^{1/2} u, g \rangle|}{\|u\|_\Sigma} \right] = B_2^{1/2} \gamma(T, \Sigma).$$

Substituting these quantities in (74) for $\|u\|_K \leq 1$ implies the result. $\qquad \square$

The following result proves a rate result for linear and logistic regression with $h(\beta) = \lambda \|\beta\|_1$ (based on the restricted strong convexity result above). Define for the loss $\ell(\cdot, \cdot)$,

$$f_n(\beta) := \frac{1}{n} \sum_{i=1}^n \ell(Y_i, X_i^\top \beta) \quad \text{and} \quad \hat{\beta} := \operatorname*{argmin}_{\beta \in \mathbb{R}^p} f_n(\beta) + \lambda \|\beta\|_1$$

Recall from Lemma 3.3 that if $\lambda = L\sigma^*(1 + 3\xi)\sqrt{2 \log(p/s)/n}$ in case of squared loss and $\lambda = (L/2)(1 + 3\xi)\sqrt{2 \log(p/s)/n}$ in case of logistic loss, for some $\xi > 0$ then on the event (41),

$$\hat{\beta} - \beta^* \in T_{\texttt{lasso}}(s(6 + 2\xi^{-1})^2) := \{\delta \in \mathbb{R}^p : \|\delta\|_1 \leq \sqrt{s} \|\delta\| (6 + 2\xi^{-1})\}.$$

This holds for both the linear and logistic lasso case.

**Proposition 3.4.** *Consider the penalized lasso estimator $\hat{\beta}$ given by*

$$\hat{\beta} := \operatorname*{argmin}_{\beta \in \mathbb{R}^p} \frac{1}{n} \sum_{i=1}^n \ell(Y_i, X_i^\top \beta) + \lambda \|\beta\|_1,$$

*where $\ell$ is either the squared or logistic loss and $\lambda$ is chosen as in Lemma 3.3 for some $\xi > 0$. Assume (A1), (A2). With $T$ defined in (22), assume that $\exists \theta > 0$ s.t. for all $u \in T$ with $\|u\|_K \leq 1$,*

$$\theta^2 \|u\|_K^2 \leq \frac{1}{n} \sum_{i=1}^n \left\{ \ell(Y_i, X_i^\top \beta^* + X_i^\top u) - \ell(Y_i, X_i^\top \beta^*) - u^\top X_i \ell'(Y_i, X_i^\top \beta^*) \right\}, \tag{23}$$

*as well as*

$$L(2 + 5\xi)\sqrt{2s \log(p/s)/n} \leq B_3^{1/2} \phi(T) \theta^2 \times \begin{cases} 1/\sigma^*, & \text{for } \ell, \text{ the squared loss,} \\ 2, & \text{for } \ell, \text{ the logistic loss.} \end{cases} \tag{24}$$

*Then with probability at least $1 - 2/(\xi^2 \log(p/s)(p/s)^\xi)$,*

$$\|\hat{\beta} - \beta^*\|_K \leq \frac{L(2 + 5\xi)}{B_3^{1/2} \phi(T) \theta^2} \sqrt{\frac{2s \log(p/s)}{n}} \times \begin{cases} \sigma^*, & \text{for } \ell, \text{ the squared loss,} \\ 0.5, & \text{for } \ell, \text{ the logistic loss.} \end{cases} \tag{25}$$

Assumption (23) is verified in Proposition F.1 and the assumptions of this proposition are satisfied for both squared and logistic loss. Proposition above proves the rate for logistic lasso with $\sqrt{\log(p/s)}$ instead of (usually seen) $\sqrt{\log(p)}$. See [15] for a similar result that requires more stringent conditions on $\Sigma$.

The argument in Proposition 3.4 is not specific to the Lasso. The same argument yields a similar bound for the Group-Lasso with rate $\sqrt{sd + \log(M/s)}/n\sqrt{n}$ under the setting of Lemma 3.5 using the Gaussian-width bound from Lemma 3.6.

*Proof of Proposition 3.4.* Set $\delta = \hat{\beta} - \beta^*$. From the definition of $\hat{\beta}$ we get that

$$
\begin{aligned}
f_n(\beta^* + \delta) + \lambda\|\beta^* + \delta\|_1 &\le f_n(\beta^*) + \lambda\|\beta^*\|_1 \\
f_n(\beta^* + \delta) - f_n(\beta^*) &\le \lambda(\|\beta^*\|_1 - \|\beta^* + \delta\|_1).
\end{aligned}
\tag{75}
$$

Adding $-\delta^\top \nabla f_n(\beta^*)$ from both sides, we get

$$
f_n(\beta^* + \delta) - f_n(\beta^*) - \delta^\top \nabla f_n(\beta^*) \le -\delta^\top \nabla f_n(\beta^*) + \lambda(\|\beta^*\|_1 - \|\beta^* + \delta\|_1).
\tag{76}
$$

Applying triangle inequality and then Hölder's inequality on $S$ (the support of $\beta^*$), we obtain

$$
\begin{aligned}
f_n(\beta^* + \delta) - f_n(\beta^*) - \delta^\top \nabla f_n(\beta^*) &\le -\delta^\top \nabla f_n(\beta^*) + \lambda(\|\delta_S\|_1 - \|\delta_{S^c}\|_1) \\
&\le -\delta^\top \nabla f_n(\beta^*) + \lambda(\sqrt{s}\|\delta\|_2 - \|\delta_{S^c}\|_1).
\end{aligned}
\tag{77}
$$

From Lemma A.1, take $Z := n^{1/2}\Sigma^{-1/2}\nabla f_n(\beta^*)$ (this will be $\tilde{Z}$ for logistic loss). Define $U = \Sigma^{1/2}Z/(L\sigma*)$ for squared loss and $U = 2\Sigma^{1/2}\tilde{Z}/L$ for logistic loss (as in the proof of Lemma 3.3), we get on event (41)

$$
f_n(\beta^* + \delta) - f_n(\beta^*) - \delta^\top \nabla f_n(\beta^*) \le \begin{cases} \sigma^* L n^{-1/2}\left[ U^\top \delta + \sigma^* L^{-1}\lambda n^{1/2}\|\delta\|\sqrt{s} \right], & \text{for squared loss} \\ 0.5 L n^{-1/2}\left[ U^\top \delta + 2L^{-1}\lambda n^{1/2}\|\delta\|\sqrt{s} \right], & \text{for logistic loss} \end{cases}
\tag{78}
$$

We will now complete the proof for squared loss and the result for logistic loss follows by replacing $\sigma^*$ by $1/2$.

$$
\begin{aligned}
f_n(\beta^* + \delta) - f_n(\beta^*) - \delta^\top \nabla f_n(\beta^*) &\le \sigma^* L \|\delta\|(2 + 5\xi)\sqrt{2s\log(p/s)/n} \\
&\le \frac{\sigma^* L \|\delta\|_K (2 + 5\xi)}{B_3^{1/2}\phi(T)}\sqrt{2s\log(p/s)/n}.
\end{aligned}
\tag{79}
$$

The last inequality above follows from the fact that $\|u\|_K \ge B_3^{-1/2}\|u\|_\Sigma \ge B_3^{-1/2}\phi(T)^{-1}\|u\|$. Now set $t = \min\{1, \|\delta\|_K^{-1}\sigma^* L(2 + 5\xi)\sqrt{s\log(p/s)/n}/(2B_3^{1/2}\phi(T)\theta^2)\}$ so that $\|t\delta\|_K \le 1$ by assumption (24). Combining (23) with $u = t\delta$ and (79) yields

$$
\begin{aligned}
\theta^2 t^2 \|\delta\|_K^2 &\le f_n(\beta^* + u) - f_n(\beta^*) - u^\top \nabla f_n(\beta^*) \\
&\le t[f_n(\beta^* + \delta) - f_n(\beta^*) - \delta^\top \nabla f_n(\beta^*)] \le t\frac{\sigma^* L \|\delta\|_K (2 + 5\xi)}{B_3^{1/2}\phi(T)}\sqrt{2s\log(p/s)/n}.
\end{aligned}
\tag{80}
$$

Rearranging this yields,

$$
t \le \frac{\sigma^* L(2 + 5\xi)}{B_3^{1/2}\phi(T)\theta^2\|\delta\|_K}\sqrt{2s\log(p/s)/n}.
$$

By definition of $t$, this inequality implies $t = 1$ and hence

$$
\frac{\sigma^* L(2 + 5\xi)}{B_3^{1/2}\phi(T)\theta^2\|\delta\|_K}\sqrt{2s\log(p/s)/n} \ge 1 \quad \Rightarrow \quad \|\delta\|_K \le \frac{\sigma^* L(2 + 5\xi)}{B_3^{1/2}\phi(T)\theta^2}\sqrt{2s\log(p/s)/n}.
$$

$\square$

# G  Proof of sparsity of $\eta$

**Proposition 3.7.** *Assume (A1), (A2). Let the setting of Lemma 3.6 be fulfilled. Fix $\lambda$ as in Lemma 3.5 for both squared and logistic loss for some $\xi > 0$ and $T$ be the cone defined in Lemma 3.5. If $\|K\|_{op} \leq C_{\max} < \infty$ and the assumptions of Proposition 3.4 hold, then*

$$\mathbb{P}\left(|\{k \in [M] : \eta_{G_k} \neq 0\}| \leq s\tilde{C}\right) \geq 1 - 2/(\xi^2 \log(M/s)(M/s)^\xi),$$

*where $\tilde{C} := 1 + C_{\max}\{2(3+\xi)(1+\xi^{-1})\}^2 B_3^2 \phi(T)^{-2}$. For the squared loss, the same holds for $\hat{\beta}$ with $\tilde{C}$ replaced by $(1+o(1))\tilde{C}$ provided $\phi(T)^{-1}\sqrt{sd + s\log(M/s)}/\sqrt{n} \to 0$.*

*Proof of Proposition 3.7.* The estimator $\eta$ is defined by

$$\eta := \underset{\beta \in \mathbb{R}^p}{\operatorname{argmin}} \frac{1}{2}\left\|K^{1/2}(\beta - \beta^*) - n^{-1/2}Z\right\|^2 + h(\beta)$$

where $h$ is the Group-Lasso penalty given by (29) for some tuning parameter $\lambda > 0$, and

$$Z := \frac{1}{\sqrt{n}}\sum_{i=1}^n K^{-1/2}X_i\ell'(Y_i, X_i^\top \beta^*).$$

For the squared loss, $Z$ is also given in (38) while $Z$ for the logistic loss is given by $\tilde{Z}$ in (39). By the KKT conditions, we get

$$0 \in K(\eta - \beta^*) - n^{-1/2}K^{1/2}Z + \partial h(\eta)$$

where $\partial h(\eta)$ is the sub-differential of $h$ at $\eta$. This implies that for any group $k$ such that $\eta_{G_k} \neq 0$,

$$\left\|\left(K(\eta - \beta^*) - n^{-1/2}K^{1/2}Z\right)_{G_k}\right\| = \lambda. \tag{81}$$

Let $\hat{A} \subset [M]$ be the set of $s$ largest values of $\|(K^{1/2}Z)_{G_k}\|$. Let $\operatorname{supp}_G(\eta) = \{k \in [M] : \eta_{G_k} \neq 0\}$ and let $\mathcal{B}$ be a subset of $\operatorname{supp}_G(\eta)$ such that $\hat{A} \cap \mathcal{B} = \emptyset$ (or equivalently $\mathcal{B} = \hat{A}^c \cap \operatorname{supp}_G(\eta)$). Note that

$$\begin{aligned}
|\operatorname{supp}_G(\eta)| &= |\operatorname{supp}_G(\eta) \cap \hat{A}| + |\operatorname{supp}_G(\eta) \cap \hat{A}^c| = |\operatorname{supp}_G(\eta) \cap \hat{A}| + |\mathcal{B}| \\
&\leq |\hat{A}| + |\mathcal{B}| = s + |\mathcal{B}|.
\end{aligned} \tag{82}$$

Hence it is enough to bound $|\mathcal{B}|$. Set $\hat{\lambda} = n^{-1/2}\max_{k \in \hat{A}^c}\|(K^{1/2}Z)_{G_k}\|$. Summing the squares of the KKT condition (81) above for $j \in \mathcal{B}$ yields

$$\begin{aligned}
|\mathcal{B}|\lambda^2 &= \sum_{k \in \mathcal{B}}\left(K(\eta - \beta^*) - n^{-1/2}K^{1/2}Z\right)_{G_k}^2 \\
&= (n^{-1/2}Z - K^{1/2}(\eta - \beta^*))^\top \left(\sum_{k \in \mathcal{B}}\sum_{j \in G_k}K^{1/2}e_je_j^\top K^{1/2}\right)(n^{-1/2}Z - K^{1/2}(\eta - \beta^*)) \\
&= (n^{-1/2}Z - K^{1/2}(\eta - \beta^*))^\top M(n^{-1/2}Z - K^{1/2}(\eta - \beta^*)),
\end{aligned} \tag{83}$$

where $M = \sum_{k \in \mathcal{B}}\sum_{j \in G_k}K^{1/2}e_je_j^\top K^{1/2}$. Taking square root and using triangle inequality, we get

$$\begin{aligned}
\sqrt{|\mathcal{B}|}\lambda &\leq \sqrt{\sum_{k \in \mathcal{B}}\|n^{-1/2}(K^{1/2}Z)_{G_k}\|^2} + \|M^{1/2}K^{1/2}(\eta - \beta^*)\| \\
&\leq \sqrt{|\mathcal{B}|}\hat{\lambda} + \|M^{1/2}K^{1/2}(\eta - \beta^*)\|.
\end{aligned} \tag{84}$$

The second inequality above follows from the fact that $\mathcal{B} \subset \hat{A}^c$ and the definition of $\hat{A}$. By (46), with probability at least given by the right hand side of (45), $\hat{\lambda} \leq (1+\xi)^{-1}\lambda$ and hence

$$\sqrt{|\mathcal{B}|} \leq (1+\xi^{-1})\|M^{1/2}K^{1/2}(\eta - \beta^*)\|/\lambda \tag{85}$$

Therefore, $\sqrt{|\mathcal{B}|} \le (1 + \xi^{-1})\|M\|_{op}^{1/2}\|K^{1/2}(\eta - \beta^*)\|/\lambda$. By strong convexity of the quadratic program (5) (cf., e.g. [6, Lemma 1] or [7, Lemma A.2]) we have

$$\tfrac{1}{2}\|K^{1/2}(\eta - \beta^*)\|^2 \le n^{-1/2}Z^T\Sigma^{1/2}(\eta - \beta^*) + \lambda\sum_{k=1}^{M}(\|\beta_{G_k}^*\| - \|\eta_{G_k}\|).$$

On event (45), by the rightmost inequality of (47) in the proof of Lemma 3.5 we have $\eta - \beta^* \in T$ for the set $T$ defined in Lemma 3.6, and by the inequalities of (47), the previous display yields

$$\tfrac{1}{2}\|K^{1/2}(\eta - \beta^*)\|^2 \le (3 + \xi)\sqrt{s}\lambda\|\eta - \beta^*\| \le (3 + \xi)\sqrt{s}\lambda\phi(T)^{-1}B_3\|K^{1/2}(\eta - \beta^*)\| \quad (86)$$

and $\|K^{1/2}(\eta - \beta^*)\| \le 2(3 + \xi)\sqrt{s}\lambda\phi(T)^{-1}B_3$. Plugging this bound back in (85) we obtain

$$\sqrt{|\mathcal{B}|} \le \sqrt{s}\,\|K_{\bar{G},\bar{G}}\|_{op}^{1/2}\,2(3 + \xi)(1 + \xi^{-1})B_3\phi(T)^{-1} \quad (87)$$

where $\bar{G} = \cup_{k\in\mathcal{B}}G_k$. Hence we obtain $|\mathcal{B}| \lesssim s$ as required for any $\mathcal{B}$ such that the ratio $\|K_{\bar{G},\bar{G}}\|_{op}^{1/2}/\phi(T)$ is bounded.

The proof for $\hat{\beta}$ (in the squared loss case) follows the same argument. The only major difference is that we have the empirical Gram matrix $\mathsf{X}^T\mathsf{X}/n$ instead of $\Sigma$ (where $\mathsf{X}$ is the design matrix with rows $X_1, ..., X_n$), and we need to bound the quantities $\|(\mathsf{X}^T\mathsf{X}/n)_{\bar{G},\bar{G}}\|_{op}$ and $\|\mathsf{X}(\hat{\beta} - \beta^*)\|/\sqrt{n}$. It is enough to notice that $\|\mathsf{X}(\hat{\beta} - \beta^*)\|/\sqrt{n} = \|\Sigma^{1/2}(\hat{\beta} - \beta^*)\|(1 + o(1))$ and $\|\mathsf{X}^T\mathsf{X}/n)_{\bar{G},\bar{G}}\|_{op} \le (1+o(1))\|\Sigma_{\bar{G},\bar{G}}\|_{op}$ by an application of [28] with the Gaussian-width bound given in Lemma 3.6. $\square$