[Reviews · NeurIPS 2019]

Reviewer 1



Simple case - squared loss and identity covariance. The key ingredient in the proof is establishing for the problem settings of interest, that the vector \hat{\beta} - \eta lie in a well behaving set T with benign Rademacher complexity. I am only giving this paper a weak accept because: (a) - The tools and techniques used in the proofs are pretty standard, and do not contain novel ideas. (b) - Even though the authors provide some motivating examples and problem settings, I am unable to imagine more general applications of their results in Machine Learning. It seems that for any new problem of interest, we will need to think from scratch to quantifying the set T and controlling its rademacher complexity. Further the estimator in (4) may not be computationally easier. I am quite flexible on this front and am willing to change my opinion based on the author feedback. (c) Writing: The paper is not very well written, and I strongly recommend the authors to refine the paper further. Some typos I observed are listed below: 1. Line 47, definition of euclidean norms. 2. Line 74, missing a K in the expression on the LHS. 3. Line 101, the Sigma should be K as per definition (9)

Reviewer 2



The main results of the paper is novel. They are stated in very general setting and can be applied to any convex estimators. But both applications on lasso and group lasso are already studied in the literature. It is unknown whether there are applications so that the theory leads to new results beyond those in the literature.

Reviewer 3



The article introduces a so-called first order expansion for regression problems. The idea is quite interesting and the results at first intriguing, but they next lead to a few major concerns: * application-wise, it seems that the results are only usable under the strong assumption of the existence of a function psi, which is possibly hard to determine (at least only simple applications are provided by the authors which seem to suggest that this is indeed the case). It would be appropriate to better discuss the applicability of the results. * technically speaking, I am a bit concerned with the third item of (A1) which seems to constrain \ell to be quadratic (see my comments below). It is not clear neither how this assumption is really "called" in the main results. Please clarify this point. * generally speaking, it is difficult to assess the limitations and width of application of the work: which problems can it handle, which not? What would be next hurdles to break? Are there definitely problematic issues for follow ups? I would like to have a clear feedback on these aspects, both technical and applied. Detailed comments: Abstract: - repetition of "Such first order expansion" (3 times) - "the Lasso", then "the lasso" Introduction: - "one observes observations"... - what does it precisely mean for "the literature of the past decade [to have] demonstrated great success"? In which context? To what avail? Which literature anyways? Please point to references. - "in terms estimation" - I do not really understand the relevance of Q1 and Q2. You're essentially trying to convince the reader here that the proposed work is useful by asking questions on relevance of the method. Q1 is clearly not really a fundamental question. Q2 is better but the real question is rather to know whether one can improve the tractability of the asymptotics of hat{beta} for instance? And there your work proposes to answer that by the first-order expansion argument. Not the other way around. - "certain smoothness assumptions" --> such as? - in passing: "certain assumptions ... implieS" - "allows transfer of results for averages to study of" --> please correct the English gramar/syntax - please specify the definition of the derivatives of \ell(.,.). This in passing imposes that \ell be properly differentiable. This is not the case for the l1 norm. How is that dealt with? - the overall setting also assumes a concentation effect of the argument of \ell as n-->infty. This is known not to be the case when n,p-->infty together for e.g., Xi~N(0,I). Thus Taylor expansions are to no avail in this case. Since it is proposed here to let p grow possibly large, it would be worth discussing what scaling for 'p' is allowed and how it relates to the data statistics. Main Results: - "There exists constants" --> exist - what is T in third display of (A1)? - the first two assumptions in (A1) feel natural, but the last one possibly stringent. First of all, isn't the denominator simply ||u||_K^2? Then, what this implies roughly speaking is that no eigenvalue of K vanishes, thus essentially that l'' is bounded below. Am I right? So it's both bounded from above and bounded from below... Isn't that equivalent to saying that only quadratic costs are allowed? This would be a major issue of course... - it would be appropriate to comment on Th2.1. What does it really say? How does it differ from prior work, what's new and interesting in it? - I do not understand in Prop2.2 why (8) holds for some B3>0. Doesn't l''(y,u) tend to 0 with u-->-infty for instance? which likely poses a problem, but I may be wrong. Or is (8) considered probabilitistically somehow assuming (A2)? In which case, it must be restated properly. What is the low... - I would restrict the title to just "Application to ..." which is more formal Application to exact risk identities - generally speaking, to be applicable, the overall work assumes that a function psi exists and can be evaluated. It would make sense to discuss this aspect in concluding remarks, say (and move the proof outline to appendix). At this point of the article, the applications considered are rather elementary, which gives a (hopefully wrong) feeling that the results of the article are only applicable to trivial scenarios.

[Author Response · NeurIPS 2019]

**Relation to prior work, novelty:** • To answer a major concern of all reviewers: we study approximation of $\hat{\beta}$ that are an order of magnitude more precise than the commonly studied risk bounds. Lasso risk bounds are of order $s \log(p/s)/n$ while our results are $(s \log(p/s)/n)^{3/2}$; similarly for group-lasso in Section 3.3. Results at such scale are not available in prior work, at the sole exception of [21, Thm 5.1] which is restricted to Lasso and squared loss, cf. lines 190-198 for comparison. • Results are such scale **are thus novel**. Proof techniques are also novel: a careful application of powerful generic chaining results from [15, 31] is needed to obtain results at this scale, cf. the theorems of Section 6 and their proofs. • Results at such finer scale are useful to characterize the risk exactly (as opposed to upper/lower bounds up to multiplicative constants), cf. Section 4; or to construct uncertainty quantification results in the form of confidence intervals, cf. Section 5. Uncertainty quantification is a major challenge in high-dimensions and calls for results at a finer scale such as those from the submission. • We invite the reviewers to revisit their scores in light of this.

**Reviewer 1:** *"(a) tools and techniques used in the proofs are pretty standard, and do not contain novel ideas."* ⇒ cf. line 1-10 above. *"(b) - Even though the authors ..., I am unable to imagine more general applications of their results in Machine Learning. (...) for any new problem of interest (...) need to think from scratch to quantifying the set T and controlling its rademacher complexity "* ⇒ Sets $T$ and their gaussian complexity have already been studied for most high-dimensional estimators by many authors: (Group-)Lasso, Slope [6], Nuclear norm [32], tensor norms, etc, see surveys [4, 16, 23, 32]. For all these examples, set $T$ already available and extension of our results to such penalty is straightforward–we'll clarify this. *"Further the estimator in (4) may not be computationally easier"*: the quantity in (4) is not an estimator since it depends on $\beta^*$; it is an approximation of $\hat{\beta}$ that can be used for applications in Secitons 4-5, including confidence intervals. *"(c) Writing: (...)"* ⇒ we'll clarify the writing and fix the typos as suggested.

**Reviewer 2:** *"main results of the paper is novel (...) applications on lasso, group lasso are already studied in literature. (...) unknown whether there are applications s.t. the theory leads to new results beyond those in literature."* ⇒ cf. line 1-10. *"The formula of the first order expansion is not computable except in some special situations. Applications that can lead to new results beyond those already in the literature will be useful to illustrate the value of such a formula."* ⇒ cf. line 18. *"In Prop 5, $T_n$ exists but its computational formula is not available. How could it be used for inference?"* ⇒ Proving $\sqrt{n}(\hat{\theta} - a^\top \beta^*) - T_n \to 0$ in probability and $T_n$ has $t$-distribution with $n$ degrees-of-freedom yields confidence intervals: $\mathbb{P}(\sqrt{n}|\hat{\theta} - a^\top \beta^*| \leq 1.96) \approx 0.95$, hence $a^\top \beta^* \in [\hat{\theta} \pm 1.96 n^{-1/2}]$ (asymptotically) with probability 0.95.

**Reviewer 3:** *"(...) results are only usable under the strong assumption of the existence of a function $\psi$ (...)"* ⇒ Our construction $\eta$ generalizes $\beta^* + \frac{1}{n} \sum_i \psi(X_i, Y_i)$ in high-dimensions, proving that such approximation exists for several $\hat{\beta}$ in line 15 above. *"technically speaking, (...) third item of (A1) which seems to constrain $\ell$ to be quadratic (see my comments below). (...) not clear how this assumption is really "called" in the main results (...)"* ⇒ $\ell$ need not be quadratic, cf. line 52-54 below. Third item of (A1) is the Restricted Strong Convexity (RSC) assumption from [32], required to obtain risk bounds for logistic lasso of order $s \log(p/s)/n$. It is used in the main theorems in Section 6 to bound certain empirical processes. The constant $B_3$ is explicit in these proofs, which allows to track where third item of (A1) is used. *"(...) difficult to assess the limitations and width of application of the work: which problems can it handle, which not? What would be next hurdles to break? Are there definitely problematic issues for follow ups?"* ⇒ cf. lines 14-17 above. *""certain smoothness assumptions" → such as?"* ⇒ Differentiability of the loss in [18,24] and stochastic equicontinuity (a weaker form of differentiability) in [35, 36]. We'll clarify this. *"please specify the definition of the derivatives of $\ell(.,.)$. This in passing imposes that $\ell$ be properly differentiable. This is not the case for the l1 norm. How is that dealt with?"* ⇒ Derivatives of $\ell(y, u)$ are always with respect to $u$. We'll clarify this. The data-fitting loss (squared, logistic) is required to be differentiable, but not the penalty $h$ ($\ell_1$-norm, ...). *"the overall setting also assumes a concentration effect of the argument of $\ell$ as $n \to \infty$. This is known not to be the case when $n, p \to \infty$ together for e.g., $X_i \sim N(0, I)$. Thus Taylor expansions are to no avail in this case. Since it is proposed here to let p grow possibly large, it would be worth discussing what scaling for 'p' is allowed and how it relates to the data statistics."* ⇒ The submission does allow for $n, p \to \infty$ together: Lasso requires $r_n \asymp (s \log(p/s)/n)^{1/2} \to 0$ cf. (23), Group-Lasso requires $r_n \asymp \{(sd + s \log(M/s))/n\}^{1/2} \to 0$, cf. Lemma 3.5. The required concentration is obtained by a careful application of powerful generic chaining results from Dirksen [15] and Mendelson [31] that let us obtain concentration results uniformly over $T$; cf. Section 6 and the corresponding proofs. *"what is T in third display of (A1)?"* ⇒ Third ineq. in (A1) is the Restricted Strong Convexity of [32], $T$ is the restricted cone. *"(...) the last [assumption in (A1)] possibly stringent. (...) isn't the denominator simply $\|u\|_K^2$? (...) this implies (...) that no eigenvalue of $K$ vanishes, thus essentially that $\ell''$ is bounded below (...) equivalent to saying that only quadratic costs are allowed? This would be a major issue"* ⇒ $\|u\|_K = \|K^{1/2} u\|$ and $K = E[\frac{1}{n} \sum_i \ell''(Y_i, X_i^T \beta^*) X_i X_i^T]$ defined in line 72 is an expected (population) quantity. $\ell''$ needs **not** be bounded below, only the population matrix $K$. For logistic loss, the assumptions hold (Prop 2.2) although $\ell''$ is not bounded from below. Third inequality in (A1) is Restricted Strong Convexity of [32], a common assumption for analysing logistic lasso/group-lasso. *"(...) appropriate to comment on Th2.1 (...) differ from prior work, what's new/interesting in it?"* ⇒ cf. line 1-10 above. *"I do not understand in Prop2.2 why (8) holds for some $B_3 > 0$. Doesn't $l''(y, u)$ tend to 0 with $u \to \infty$ for instance?"* ⇒ Third ineq in (8) involves $K$ and $\Sigma$ which are both expectations. $\ell(y, u) \to 0$ as $u \to \infty$ is OK as long as $\|\Sigma^{1/2} \beta^*\| \leq 1$ (or $\leq C$) (cf line 109). This is common assumption in logistic lasso, e.g. Prop 6.2 in [1]; though our proof is not restricted to Gaussian $X_i$–we'll clarify.

[Meta-Review · NeurIPS 2019]

The present paper proposes an approximation, based on the first order Taylor expansion of convex regularizer. In the regularized regression setting and under some mild condition on the loss function and the underlying distribution that generates the data, the authors prove that one can replace the regularization term of the regression algorithm by its Taylor approximation and have a guarantee that the solution obtain with this approximation will be close to the original solution (according to the Mahalanobis distance). The authors give then examples of such proxy for square loss and logistic regression and also for Constrained Lasso, Penalized Lasso and Group Lasso. The paper also proposes a discussion where this approach can be useful. Although this paper is a bit technical, it is well written and the result are on my opinion non trivial and interesting. The reviewers points out that the approach needs to the user to define some set $T$ with precise properties in order to make this approach working for a particular regression algorithm. In fact it has been the principal weakness raised by the reviewers. On my opinion, this is indeed a problem, but not big enough to prevent acceptation. Logistic regression, Lasso and Group Lasso are important enough to justify the interest of the approach. Moreover, in the rebuttal, the authors pointed out that this set $T$ has already been used in other situations in the literature: “Sets T and their gaussian complexity have already been studied for most high-dimensional estimators by many authors: (Group-)Lasso, Slope [6], Nuclear norm [32], tensor norms, etc, see surveys [4, 16, 23, 32]. For all these examples, set T is already available and extension of our results to such penalty straightforward–we’ll clarify this.” So I decide not to take too much account of this “weakness” in my final decision. Indeed, I consider the results contain in this paper as non –trivial and interesting for the supervised regression community, and I therefore recommend its acceptation.